# Developmental estrogen exposure in mice disrupts uterine epithelial cell differentiation and causes adenocarcinoma via Wnt/β-catenin and PI3K/AKT signaling

Elizabeth Padilla-Banks[1☯], Wendy N. Jefferson[1☯], Brian N. Papas[2☯], Alisa A. Suen[1], Xin Xu[3], Diana V. Carreon[1], Cynthia J. Willson[4], Erin M. Quist[5¤], Carmen J. Williams[1]*

**1** Reproductive and Developmental Biology Laboratory, National Institute of Environmental Health Sciences, National Institutes of Health, Research Triangle Park, North Carolina, United States of America, **2** Integrative Bioinformatics, Biostatistics and Computational Biology Branch, National Institute of Environmental Health Sciences, National Institutes of Health, Research Triangle Park, North Carolina, United States of America, **3** Epigenetics and Stem Cell Biology Laboratory, National Institute of Environmental Health Sciences, National Institutes of Health, Research Triangle Park, North Carolina, United States of America, **4** Inotiv-RTP, Research Triangle Park, North Carolina, United States of America, **5** Experimental Pathology Laboratories, Research Triangle Park, North Carolina, United States of America

☯ These authors contributed equally to this work.
¤ Current address: Charles River Laboratories, Durham, North Carolina, United States of America
* williamsc5@niehs.nih.gov

**Data Availability Statement:** The sequencing data generated in this study were deposited in the Gene

## Abstract

Tissue development entails genetically programmed differentiation of immature cell types to mature, fully differentiated cells. Exposure during development to non-mutagenic environmental factors can contribute to cancer risk, but the underlying mechanisms are not understood. We used a mouse model of endometrial adenocarcinoma that results from brief developmental exposure to an estrogenic chemical, diethylstilbestrol (DES), to determine causative factors. Single-cell RNA sequencing (scRNAseq) and spatial transcriptomics of adult control uteri revealed novel markers of uterine epithelial stem cells (EpSCs), identified distinct luminal and glandular progenitor cell (PC) populations, and defined glandular and luminal epithelium (LE) cell differentiation trajectories. Neonatal DES exposure disrupted uterine epithelial cell differentiation, resulting in a failure to generate an EpSC population or distinguishable glandular and luminal progenitors or mature cells. Instead, the DES-exposed epithelial cells were characterized by a single proliferating PC population and widespread activation of Wnt/β-catenin signaling. The underlying endometrial stromal cells had dramatic increases in inflammatory signaling pathways and oxidative stress. Together, these changes activated phosphoinositide 3-kinase/AKT serine-threonine kinase signaling and malignant transformation of cells that were marked by phospho-AKT and the cancer-associated protein olfactomedin 4. Here, we defined a mechanistic pathway from developmental exposure to an endocrine disrupting chemical to the development of adult-onset cancer. These findings provide an explanation for how human cancers, which are often associated with abnormal activation of PI3K/AKT signaling, could result from exposure to environmental insults during development.

Expression Omnibus database under accession code GSE218156 and are publicly available.

**Funding:** This research was supported by the Intramural Research Program of the NIH, National Institute of Environmental Health Sciences, 1ZIAES102405 (CJW). The funder had no role in study design, data collection and analysis, decision to publish, or preparation of the manuscript.

**Competing interests:** The authors have declared that no competing interests exist.

**Abbreviations:** BSA, bovine serum albumin; cKO, conditional knockout; CO, control; DEG, differentially expressed gene; DES, diethylstilbestrol; DevGE, developing glandular epithelium; ERα, estrogen receptor alpha; EpSC, epithelial stem cell; FBS, fetal bovine serum; GE, glandular epithelium; GPC, glandular progenitor cell; GO, Gene Ontology; HE, hematoxylin & eosin; IEC, intestinal epithelial cell; IHC, immunohistochemistry; LE, luminal epithelium; LPC, luminal progenitor cell; MGE, mature glandular epithelium; NIEHS, National Institute of Environmental Health Sciences; PC, progenitor cell; scRNAseq, single-cell RNA sequencing; UMAP, uniform manifold approximation and projection.

## Introduction

Cellular differentiation is genetically and epigenetically programmed but strongly influenced by the environment. Environmental cues provide opportunities for the developing organism or differentiating adult tissues to adapt their pre-programmed differentiation trajectory to improve fitness [1,2]. Altered developmental trajectories can also reduce adult fitness, particularly in settings where the environment changes after there are developmental adaptations to the initial environment. This phenomenon, now termed "developmental origins of health and disease," was first discovered as a connection between poor nutrition in infants and increased risk of arteriosclerotic heart disease in prosperous adults [3,4].

Environmental exposures during specific windows of development can even lead to delayed onset of cancer in adulthood. A classic example of this phenomenon is prenatal exposure to diethylstilbestrol (DES), which significantly alters developmental patterning of the female reproductive tract. This altered patterning leads to adult reproductive tract structural defects and functional deficits including infertility, miscarriage, and preterm birth [5]. Fetal DES exposure is also associated with an increased incidence of specific cancers in adult women [6–8]. The "two-hit hypothesis" of cancer development postulates that 2 mutational events cause cancer, with the first mutation causing cancer susceptibility and the second mutation causing cancer progression [9–11]. Indeed, numerous cancers result from 2 mutational events, such as retinoblastoma and colorectal carcinoma [12,13]. In addition to mutations, epigenetic alterations can similarly serve as "hits" responsible for cancer development through their impacts on tumor suppressors or oncogenes [14].

How do non-mutagenic environmental exposures during development impact tissues in a way that leads to late onset of carcinogenesis, particularly when similar exposures during adulthood have no persistent effects? To address this question, we are utilizing a mouse model of DES exposure during female reproductive tract differentiation that was initially developed to model human DES exposure [15]. At birth, the mouse female reproductive tract is a bifurcated tubular structure lined by a simple epithelium. The uterus is not fully differentiated until about 3 weeks later, when anterior-posterior patterning and cellular differentiation are fully established under the influence of factors including *Hox* and *Wnt* genes, growth factors, Hippo signaling, and steroid hormones [16,17]. The mature uterine endometrium has a luminal epithelium and a glandular epithelium formed from invaginations of the luminal epithelium into the underlying stroma. These epithelia undergo cyclic regeneration with each estrous cycle from rare adult epithelial stem cells (EpSCs), which provide progenitor cells (PCs) committed to proliferate and differentiate into luminal or glandular epithelium [18–21]. The presence of uterine EpSCs was previously described using lineage tracing and flow cytometry experiments; however, their location and specific cell markers are inconclusive [19]. In addition, previous uterine single-cell RNA-seq was performed on either neonatal/prepubertal mice or had shallow sequencing, so stem cell populations were not adequately captured [17,22,23].

The mouse model of neonatal DES exposure entails exposing newborn female mice to DES daily for 5 days beginning on the day of birth, during the initial window of rapid reproductive tract differentiation. This exposure causes altered reproductive tract patterning, increased deposition of extracellular matrix material throughout the tissue, and a high incidence of estrogen receptor alpha (ERα)-dependent uterine endometrioid adenocarcinoma in 12-month-old adults [15,24,25]. This model requires 2 "hits": neonatal exposure to an estrogenic chemical followed by additional exposure to endogenous estrogen following puberty. The resulting cancers have no obvious mutational signature and are not discrete tumors [15,26–28]. Instead, they are characterized by having multiple foci of diffuse infiltrating neoplastic cells with mixed glandular, basal, and squamous cell features [15,25].

Basal epithelial cells in the female reproductive tract are normally restricted to the cervix and vagina [25,29]. In vaginal epithelium, basal cells form the first cell layer that abuts the basement membrane between stromal and epithelial cells. These basal cells have high expression of SIX1, a homeobox transcription factor and human oncogene [30]. Following neonatal DES exposure, basal cells expressing SIX1 are also found in many regions of the uterine body and uterine horns at the basement membrane beneath endometrial luminal and glandular epithelium (GE) cells [25]. SIX1 also marks neonatal DES exposure-induced cancer cells and is found in some human endometrial cancers [29]. Conditional deletion of SIX1 in the mouse uterus prevents uterine basal cell formation following DES exposure but does not prevent cancer, indicating that neither SIX1 nor basal cells are required for cancer development in this model [31].

To determine the mechanisms underlying neonatal DES exposure-induced cancer development, we utilized single-cell RNA sequencing (scRNAseq) and spatial transcriptomics [32] to identify alterations in uterine gene expression in adult DES-exposed mice with adenocarcinoma relative to unexposed controls. We found that DES-exposed mice lack normally differentiated uterine luminal and glandular epithelial cells. Instead, they have an expanded population of less differentiated uterine epithelial cells that have characteristics of stem and progenitor cells and extensive activation of Wnt/β-catenin signaling. The endometrial stroma is highly enriched in inflammatory pathways, and the cancer cells have activated PI3K/AKT signaling, a common driver of uterine adenocarcinoma. Our findings suggest that brief estrogenic chemical disruption of uterine epithelial cell differentiation, combined with stromal inflammation, promotes abnormal activation of Wnt/β-catenin and PI3K/AKT signaling pathways that drive cancer development.

## Results

### Identification of uterine cell types in control and DES-exposed mice

Single cells were isolated from a single sample of pooled uteri of control (CO) or DES-exposed mice, processed using the 10x Genomics scRNAseq platform, and sequenced to a depth of >1.2 billion reads per sample with >40,000 reads per cell. Mapping was of high quality and 32,387 CO cells and 16,312 DES cells had sufficient information captured (S1A Table). There was a robust number of transcripts per cell (CO, 6779; DES, 10153). Unsupervised clustering of all cells was performed using Seurat v3.1.0 and a uniform manifold approximation and projection (UMAP) generated (Fig 1A). Clustered cell types were identified by comparing differentially expressed genes (DEGs) of each cluster to published uterine cell type markers [22] (Fig 1B and S1B Table). The largest clusters were epithelial and mesothelial cells, but there was representation of most expected cell types. Three cell types normally found in the uterine stromal region were largely absent from the DES sample: stromal cells, pericytes, and endothelial cells.

The lack of stromal cells in the DES sample was surprising compared to the robust capture of CO stromal cells. One explanation for this finding was that another cell type in the DES sample was not categorized properly. Indeed, there was a group of DES cells that categorized as epithelial cells but plotted closer on the UMAP to CO stromal cells and mesothelial cells than to CO epithelial cells. To test for mis-categorization, we first performed unbiased clustering of DES mesothelial cell gene expression with CO mesothelial and stromal cell gene expression. There was no resulting subset of DES mesothelial cells that had characteristics of stromal cells (S1A Fig). Next, a comparison of DES epithelial cell gene expression to both CO epithelial and stromal cells revealed a subcluster of DES cells that exhibited characteristics of both cell types (Fig 1C). Of note, most of the epithelial markers appeared to be expressed at a lower level

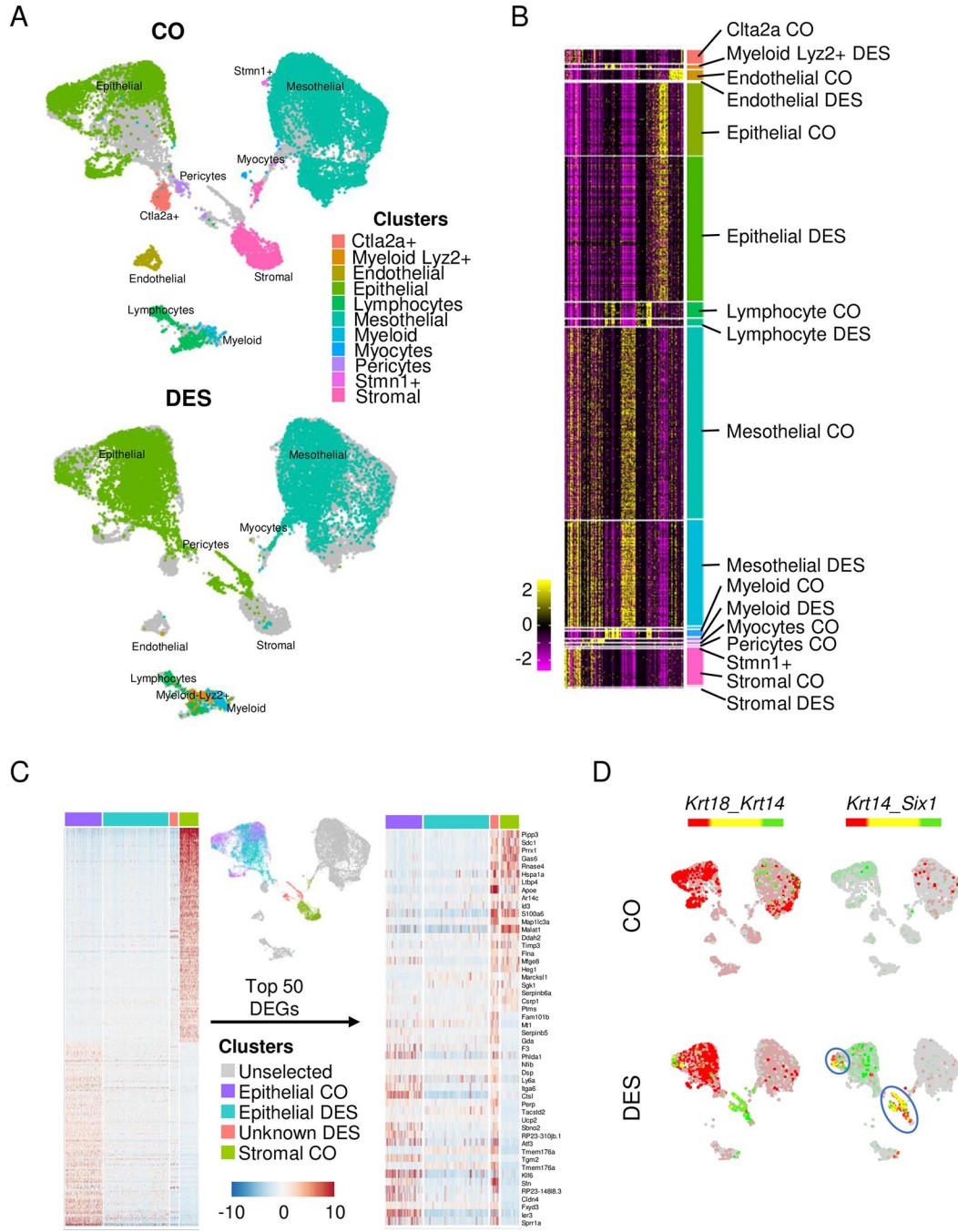

**Fig 1. Identification of uterine cell types captured by scRNAseq.** (A) Integrated UMAP of all uterine cells captured. Each cell type identified by color and text from CO (top) and DES (bottom). (B) Heat map of gene expression of cell identity genes. Cell type indicated for each identified cluster. Expression is centered [mean = 0 ± SD of each feature]. (C) Heat map of epithelial and stromal cell markers in CO and DES epithelial cells, CO stromal cells and an unidentified population with similarities to both cell types (left). Top 25 cell identity DEGs are plotted for both epithelial and stromal cells (50 total). Expression is Pearson Residuals from the SCTransform method. (D) Dual feature plots of epithelial and basal cell markers (*Krt18*, *Krt14*, and *Six1*). The blue circles indicate cells with a basal cell signature. CO (top) and DES (bottom). In this and all subsequent dual feature plots, colors for each gene (red or green) are indicated above the UMAPs; yellow indicates overlapping expression. The data underlying this figure can be found in the Gene Expression Omnibus database under accession code GSE218156. CO, control; DEG, differentially expressed gene; DES, diethylstilbestrol; scRNAseq, single-cell RNA sequencing; SD, standard deviation; UMAP, uniform manifold approximation and projection.

in DES epithelial cells compared to CO epithelial cells. The unidentified DES subcluster was confirmed by a heat map of the top 50 DEGs differentiating CO epithelial from CO stromal cells (Fig 1C). DEGs in this DES subcluster included the basal cell markers *Trp63* and *Krt14*. Neonatal DES exposure causes development of a population of basal cells, not observed in controls, that co-express *Trp63*, *Krt14*, and *Six1* [25,29,31]. Feature plots of the epithelial cell marker, *Krt18*, and these basal cell markers clearly identified 2 DES cell clusters as basal cells. One of these clusters was near the CO stromal cells and one was adjacent to the main epithelial cell cluster; this cell type was not observed in CO cells (Figs 1D and S1B). These findings confirm that basal cells were included in the analyzed DES cell populations. Feature plots of 3 stromal cell markers, *Dpt*, *Vcan*, and *Col6a3* [23], confirmed that the DES group lacked stromal cells (S1C Fig). It is likely that increased levels of extracellular matrix deposition in the stroma of DES uteri precluded isolation of living stromal cell types for analysis [15,33].

## Comparison of CO and DES stromal cells using spatial transcriptomics

Because DES stromal cells were not captured in our scRNAseq analysis, we used spatial transcriptomics to characterize gene expression in DES stromal cells. Frozen longitudinal sections from uteri of 12-month-old CO and DES mice were evaluated to ensure inclusion of the luminal region. DES sections were also evaluated to determine the presence of uterine adenocarcinoma. Once appropriate regions were confirmed in 2 tissue blocks per group (CO A and B; DES A and B), immediately adjacent sections were processed for spatial transcriptomics (Figs 2A and S2A). There were approximately 40 million total reads per sample. One of the CO sections (CO B) had a substantial tissue fold and one of the DES sections (DES B) was limited in identifying clusters, so only 1 section per group was chosen for the majority of the analyses. However, some analyses were performed on these additional sections to confirm reproducibility in findings (S2A Fig). All CO and DES samples had similar numbers of spots under the tissue (1,357 to 1,834), levels of sequencing saturation (56% to 66%), and median genes per spot (2,056 to 2,518) (S2A Table). Note that each spot, which is 55 μm in diameter (10x Genomics), represents gene expression from approximately 5 to 10 cells, depending on cell size. For this reason, clear distinctions in gene expression cannot be made for intermixed cell types using this methodology.

Spot clusters were distinguished in all tissue sections using Space Ranger-1.2.2, skmeans 10 (10x Genomics) (Figs 2A and S2A and S2B–S2E Table). Cluster cell type identification was made using published markers (Figs 2B and S2B) [22,23,25,34,35]. CO A sections had clusters identifiable as regions of muscle, stroma, and epithelium (Fig 2A and 2B). Glandular epithelium was present in CO A cluster 5 based on having high expression of *Foxa2* and *Sult1d1*. CO A cluster 10 could not be characterized as one specific cell type and likely represented a mixture of epithelial and stromal cells in close proximity. CO B sections also had identifiable regions of muscle, epithelium, and stroma, with the most clearly identified stroma in cluster 6 (S2A and S2B Fig). The DES A clusters had regions of muscle, stroma, and regions representing a mixture of epithelial and basal cells (Fig 2A and 2B). DES B clusters had similar identities (S2A and S2B Fig). Cell type identification was confirmed using violin plots of a highly expressed marker for each cell type and/or spatial expression plots showing their histologic locations (Figs 2C and S2C). Stromal cells were identified clearly in DES A cluster 4 and DES B cluster 5.

To explore gene expression differences between CO and DES stromal cells, we overlapped the 431 DEGs in CO A cluster 8 (S2B Table) with 468 DEGs in DES A cluster 4 (S2C Table). Only about 30% of the DEGs were in common, including stromal cell markers *Vcan*, *Dpt*, and *Col6a3* (Fig 2D and S3A–S3C Table). Interestingly, the well-characterized stromal cell marker,

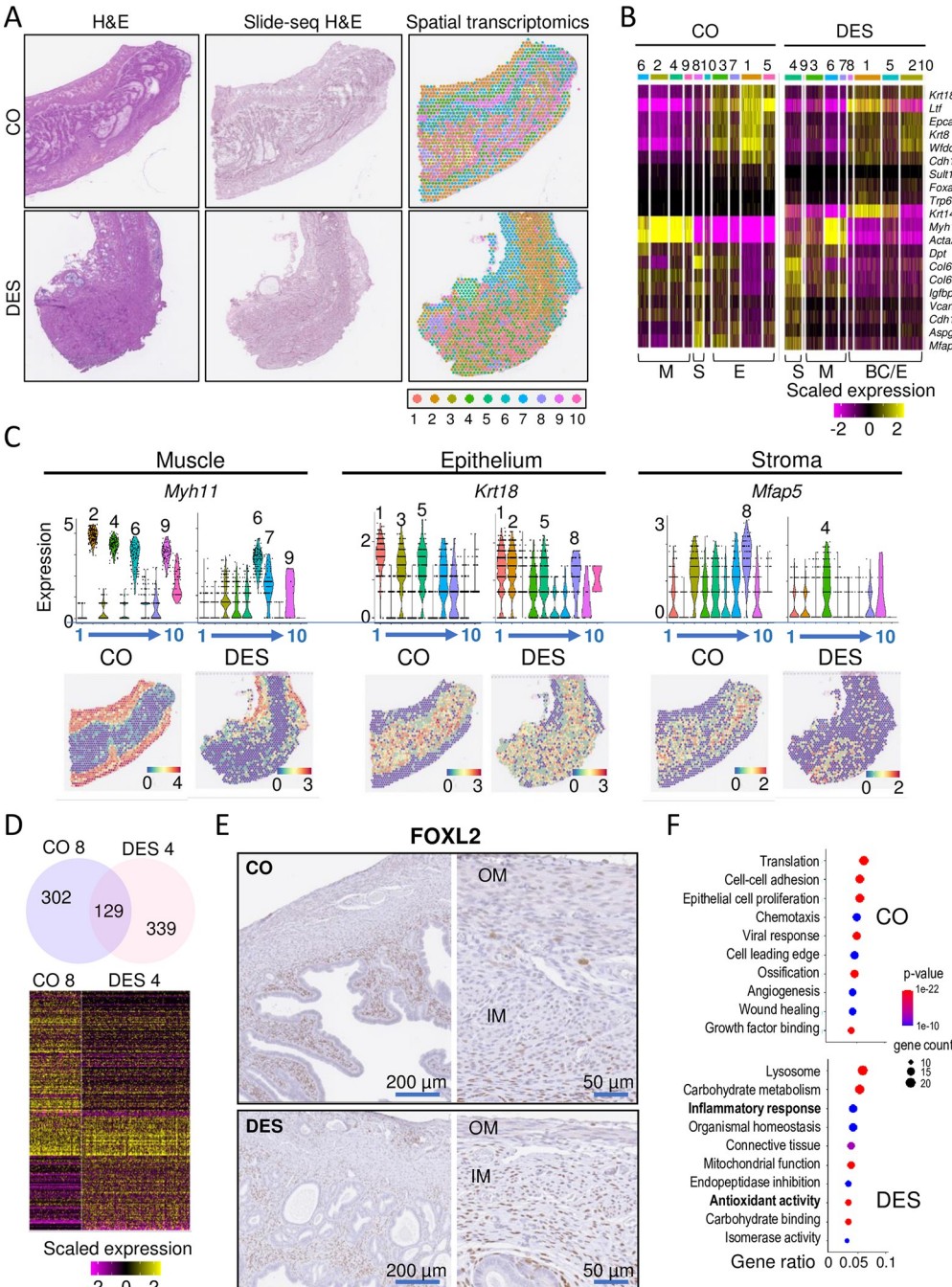

**Fig 2. Stromal cell gene expression is altered by DES as evidenced by spatial transcriptomics.** (A) Uterine tissue sections from CO A and DES A used for ST. HE stain of tissue section adjacent to ST section (left) and tissue sections used for ST (Slide-seq HE, middle). Cluster identification using Space Ranger-1.2.2, skmeans 10 (10x Genomics) (right); colors represent distinct clusters. (B) Heat map of select uterine tissue cell type markers plotted for CO A (left) and DES A (right). Cell type is indicated below heat maps (M = muscle, S = stroma, E = epithelium, BC = basal cells). Expression values are Pearson Residuals from the SCTransform method. (C) Violin plots and ST section of select markers for CO A and DES A (cluster number indicated below plot). Violin plot expression same as panel B; ST expression = natural log transformed counts. (D) Venn diagram of CO A and DES A stromal cluster specific DEGs (top). Heat map of CO A and DES A stromal cell markers (bottom). Expression same as panel B. (E) Representative FOXL2 IHC in 12-month-old CO- and DES-exposed uteri ($n$ = 4–6 mice per group). CO (top) and DES (bottom); right panels are higher magnification. OM, outer muscle; IM, inner muscle. (F) Top 10 non-overlapped GO categories for CO (top) and DES (bottom). Gene ratio (#genes in GO category/#DEGs), $p$-value, and gene count indicated. The data underlying this figure can be found in the Gene Expression Omnibus database under accession code GSE218156.

CO, control; DEG, differentially expressed gene; DES, diethylstilbestrol; GO, Gene Ontology; HE, hematoxylin & eosin; IHC, immunohistochemistry; ST, spatial transcriptomics.

*Foxl2* [35], was a DEG in CO A but not DES A stromal cell clusters. A comparison of stromal cell clusters for CO B (cluster 6) and DES B (cluster 5) revealed even less overlap, with only 14% of the DEGs in common (S3H–S3J Table). Immunohistochemical staining of FOXL2 in CO sections revealed strong staining in the inner stroma adjacent to epithelium and weaker staining in the outer stroma and muscle layers (Figs 2E and S2E). In DES uterine sections, FOXL2 staining was similar across stroma and muscle, explaining why *Foxl2* was not a DEG for DES stroma.

The most significantly enriched biological processes identified by Gene Ontology (GO) analysis of CO A-specific stromal DEGs included extracellular matrix organization, translation, and regulation of signal transduction (S3D Table). GO analysis of DES A-specific stromal DEGs also identified extracellular matrix organization, but unlike the GO categories for CO DEGs, oxidative phosphorylation and glycolytic process were identified as significantly enriched (S3E Table). The top non-overlapping GO categories revealed substantial differences between CO A and DES A (Fig 2F and S3F and S3G Table). GO analysis of CO B and DES B stromal DEGs resulted in >70% overlap with highly enriched GO categories of the corresponding treatment group (S2D Fig and S3K–S3N Table), providing consistency in spatial transcriptomics results across independent samples. Relevant to cancer development, both DES A and B stromal cells were particularly enriched in categories related to antioxidant activity and inflammatory responses.

## DES epithelial cells lack luminal or glandular identity

To determine how epithelial cell types differed between CO and DES samples, we restricted our scRNAseq analysis to include only cells differentially expressing the epithelial cell marker *Krt18*. Integrated UMAP analysis identified 4 distinct cluster regions with almost no overlap between CO and DES cells (Fig 3A). To determine which cell types were present in each cluster, marker genes for each cluster were determined by Seurat+MAST (S3A Fig and S4 Table). Two DES-specific clusters (8 and 11) separate from the main grouping of DES cells differentially expressed 2 basal cell markers (*Krt14* and *Trp63*) and *Six1*, which is mainly expressed in basal cells (Fig 3A and S4 Table) [25]. Dual feature plots confirmed the identity of these 2 DES clusters as basal cells based on their high, overlapping expression of *Krt14* and *Trp63* (Fig 3A). As expected, CO epithelial cell clusters lacked the basal cell markers.

A group of 3 CO-specific clusters (13, 17, and 19) were separate from the largest grouping of CO cells (Figs 3A and S3A). These clusters differentially expressed several uterine gland-specific genes, including *Prss29*, *Spink1*, *Sult1d1*, *Napsa*, *Gpx3*, and *Klk1*, indicating that they represented glandular epithelium (GE) (S3A Fig and S4 Table) [17,36]. Of these markers, *Prss29* and *Spink1* are highly expressed in mature GE, whereas *Napsa* and *Klk1* are expressed in developing GE. Dual feature plots of *Klk1* and *Prss29* showed non-overlapping expression in these clusters with *Klk1* expressed in clusters 13 and 17 and *Prss29* expression in cluster 19, suggesting clusters 13 and 17 were developing GE and cluster 19 was mature GE (Fig 3A). The large grouping of CO epithelial cell clusters (0, 1, 14, 15) were presumed luminal epithelium (LE) cells as they comprised the largest number of epithelial cells in the mouse uterus (Fig 3A). The large grouping of DES epithelial cells did not overlap with either the presumed CO LE cells or the CO GE though there were a few CO cells clustered with this DES group (Figs 3A and S3A and S4 Table).

The lack of overlap in UMAP locations of the CO and DES epithelial cells indicated that these populations were quite different in their gene expression and precluded further joint

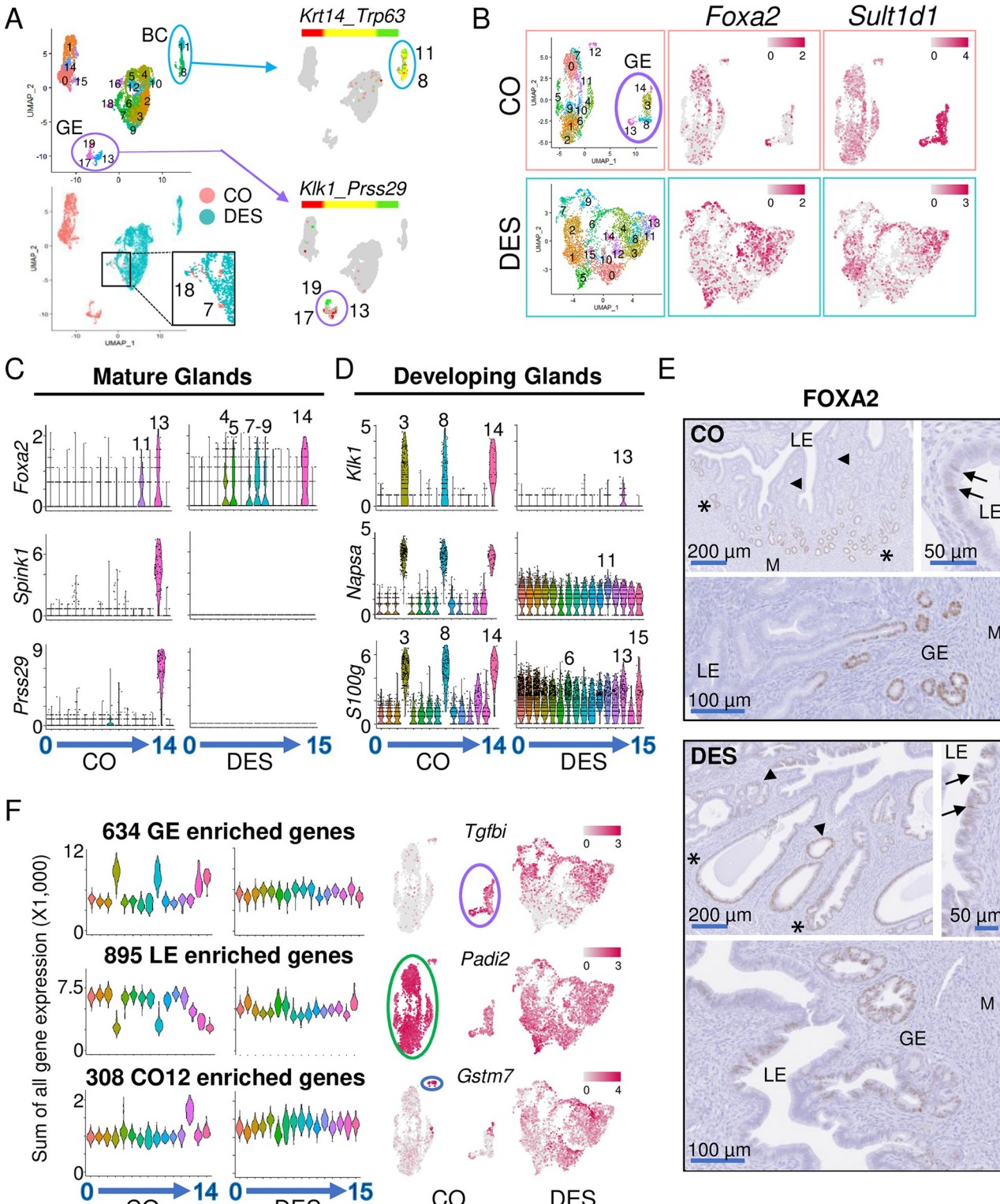

**Fig 3. Epithelial cells from DES-exposed mice lack subtype identity.** (A) UMAP of integrated epithelial cells from CO and DES scRNA-seq samples (top left); cluster numbers indicated. Epithelial cells identified as either CO or DES on the integrated UMAP (bottom left). Overlapping CO and DES cells found in only 1 region (clusters 7 and 18, magnified area indicated). Dual feature plots (right) for BC markers (*Krt14* and *Trp63*) and GE markers (*Klk1* and *Prss29*). (B) UMAPs of non-integrated CO and DES epithelial cells. GE clusters outlined in purple. Feature plots of *Foxa2* and *Sult1d1* for CO (top) and DES (bottom). (C) Violin plots of mature GE markers for CO and DES; cluster numbers on x-axes. Expression levels indicated on y-axes

as natural log transformed counts. High expression cluster numbers indicated. (D) Violin plots of developing GE markers for CO and DES; cluster numbers on x-axes. Expression levels indicated on y-axes as natural log transformed counts. High expression cluster numbers indicated. (E) Representative FOXA2 IHC in CO (top) and DES (bottom) (*n* = 4–6 mice per group). LE, luminal epithelium; GE, glandular epithelium; M, myometrium; asterisks, outer GE; arrowheads, inner GE; arrows, LE FOXA2 expression. (F) Violin plots of summed expression (SCTransform counts X 1,000) of GE, LE, and CO cluster 12 (CO12) specific DEGs for CO and DES epithelial cell clusters. Cluster numbers on x-axes. Feature plots of a representative gene from each category (right). Expression levels are indicated on all feature plots. The data underlying this figure can be found in the Gene Expression Omnibus database under accession code GSE218156. BC, basal cell; CO, control; DEG, differentially expressed gene; DES, diethylstilbestrol; GE, glandular epithelium; IHC, immunohistochemistry; LE, luminal epithelium; scRNAseq, single-cell RNA sequencing; UMAP, uniform manifold approximation and projection.

analysis of subclusters. Instead, we used a non-integrated approach to identify subclusters in CO and DES epithelial cells independently. We first removed the basal cells from the DES epithelial cells based on their expression of both *Krt14*>1 count and *Trp63*>1 count and then performed a separate UMAP analysis of the remaining epithelial cells in each group. This analysis identified 15 CO clusters and 16 DES clusters (Fig 3B and S5A and S5B Table). The CO clusters contained 2,629 cells with 3 distinct groups of cells: a small single cluster 12; a group containing clusters 14, 3, 8, and 13; and a large group containing all other clusters. The DES epithelial clusters contained 4,610 cells that were not clearly separated into groups, indicating that they had less differential gene expression between clusters. Feature plots of the mature GE marker, *Foxa2*, showed strong expression in CO cluster 13, whereas the general GE marker, *Sult1d1*, was strongly expressed in CO clusters 3, 8, 13, and 14, indicating that this group of 4 CO clusters was the GE cells (Fig 3B). In the DES cells, feature plots of *Foxa2* and *Sult1d1* showed enriched expression of both in multiple clusters but no distinctly identifiable GE clusters (Fig 3B).

To further confirm the identity of the putative CO GE clusters and to identify GE in DES clusters, violin plots were generated for genes expressed in mature GE cells including *Foxa2*, *Spink1*, and *Prss29* [36] (Fig 3C). CO cluster 13 highly expressed all 3 markers, indicating that this cluster contained the mature GE cells. *Foxa2* was also expressed in CO cluster 11 within the putative LE cell grouping; this cluster lacked expression of *Spink1* and *Prss29*, indicating that these cells were not mature GE. In the DES cells, *Foxa2* was highly expressed in multiple DES clusters but there was no corresponding expression of *Spink1* or *Prss29*, indicating that none of these clusters were mature GE. The close proximity of CO clusters 3, 8, and 14 to the *Foxa2*+ mature GE CO cluster 13 suggested that these clusters might represent GE in an earlier stage of differentiation. To test this idea, we generated violin plots of developing GE markers [17] (Fig 3D). High expression of *Klk1* and *Napsa* in CO clusters 3, 8, and 14, but not 13, confirmed these clusters as developing GE. Several other genes exhibited the same expression pattern, including *S100g*, *Hif1a*, *Prap1*, and *Wnt7b* (Figs 3D and S3B). In the DES clusters, these markers were quite evenly distributed among almost all clusters, precluding identification of developing GE clusters (Figs 3D and S3B).

To confirm the identification of CO clusters as developing and mature GE and LE, we examined FOXA2 localization in uterine sections (Figs 3E and S3C). In CO uteri, FOXA2 was primarily found in GE; however, not all GE expressed FOXA2. FOXA2 staining was highest in the distal GE (farthest away from LE) and was not detected in invaginating GE or GE nearest the LE, consistent with previous findings [37]. Some luminal cells also expressed FOXA2, likely representing CO cluster 11 and supporting the identification of the large grouping of CO clusters as LE; sporadic LE expression of FOXA2 has been described previously (Figs 3C, 3E and S3C) [37]. In DES uteri, FOXA2 was sporadically expressed in GE with no distinct pattern with respect to proximity to LE and was expressed in some areas of invaginating epithelium. There were many more FOXA2+ LE cells compared to the number in CO uteri. These findings indicated that some *Foxa2*+ DES clusters were GE but that they did not exhibit the normal GE gene expression patterns observed in controls.

Our identification of the CO GE cells strongly suggested that the remaining CO clusters found in the largest group were LE cells. To identify genes that were relatively specific to these CO putative LE clusters with low or no expression in GE, we selected DEGs expressed in at least 75% of cells in the LE clusters and no more than 15% of cells in the GE clusters; CO cluster 12 DEGs were omitted due to its independent clustering. This analysis resulted in 8 genes that were relatively LE-specific: *Ifi203*, *Pla2g2e*, *Itgam*, *Cdc42ep2*, *Lrrc26*, *Lrmp*, *Cyp21a1*, and *Adgrg7*. Dot plots of these 8 genes showed robust expression in CO LE clusters, reduced expression in all 4 of the CO GE clusters and variable expression in CO cluster 12 (S3D Fig). The DES clusters exhibited a high degree of variability in expression of these 8 genes, with no clusters enriched in more than 4 of the genes, precluding identification of DES LE clusters. Feature plots of both integrated and non-integrated UMAPs of a representative LE gene, *Adgrg7*, confirmed robust specific expression of this gene in CO LE cells but evenly dispersed expression across most DES cells (S3D Fig).

We next took a global approach to distinguish LE and GE-specific gene expression in the DES clusters. The CO clusters were merged into 3 groups: (1) CO GE clusters (3, 8, 13, and 14); (2) CO LE clusters (0, 1, 2, 4, 5, 6, 7, 9, 10, and 11); and (3) CO cluster 12. Genes that were increased 1.2-fold in 1 group over both other groups with padj<0.01 were considered "specific" to that group (S6A–S6C Table). This procedure identified 634 GE-specific genes, 895 LE-specific genes, and 308 CO cluster 12-specific genes. Violin plots of the sum of the gene expression in each category (GE, LE, or CO12) observed in each individual CO cluster showed robust differential gene expression and easily identifiable patterns (Fig 3F). In DES clusters, this cell type-specific gene expression was not observed in summary violin plots of the cell type-specific genes or the top 5 DEGs in each cell type category (Figs 3C, 3D, 3F and S3E). Feature plots of an example gene specific to each category confirmed the cell type-specific patterns in CO clusters; however, no specificity was observed in DES clusters (Fig 3F). These data demonstrate that DES epithelial cells lack cell type-specific gene expression.

## Differentiation trajectory of glandular epithelium in controls

The separation of CO GE into 4 clusters along with markers of mature GE in 1 cluster and developing GE markers in the other 3 suggested the GE clusters were at different stages of differentiation. To explore this idea further, we re-clustered only the 4 CO GE cell clusters followed by a Slingshot analysis [38] (Fig 4A). Ten resulting Slingshot clusters had 1,205 up-regulated DEGs that represented only 1 trajectory pattern (Fig 4A and S7A Table). The starting point was selected using PC markers *Foxm1* and *Top2a* that were found in Slingshot cluster 6. Developing GE markers *Klk1* and *Napsa* were observed in Slingshot clusters 0, 1, and 2, and mature GE markers *Foxa2*, *Spink1*, and *Prss29* in Slingshot clusters 5 and 7 [17,36,39,40]. These markers confirmed the trajectory pattern and the direction of developmental gene expression.

To identify more precisely markers specific to different stages of GE development, we first restricted the Slingshot cluster genes to GE-specific markers by excluding those also found in LE cell clusters, such as *Top2a* and *Foxa2*. Overlapping the 1,205 up-regulated Slingshot cluster genes with 635 GE-specific up-regulated genes resulted in 170 GE-specific genes that contributed to the trajectory of GE cell development (Fig 4B and S7B Table). A feature plot of one of these genes, *Klk1*, demonstrated specificity to the developing GE clusters identified previously (Fig 4B). Pseudotime plots of 6 of the most highly differentially expressed GE-specific genes across the Slingshot clusters demonstrated progression through differentiation (Fig 4C). This progression is summarized in a circle of differentiation time mapped back onto the non-integrated map of CO GE clusters (Fig 4D). These data confirmed that CO GE cluster 14 was the glandular progenitor cell (GPC) population.

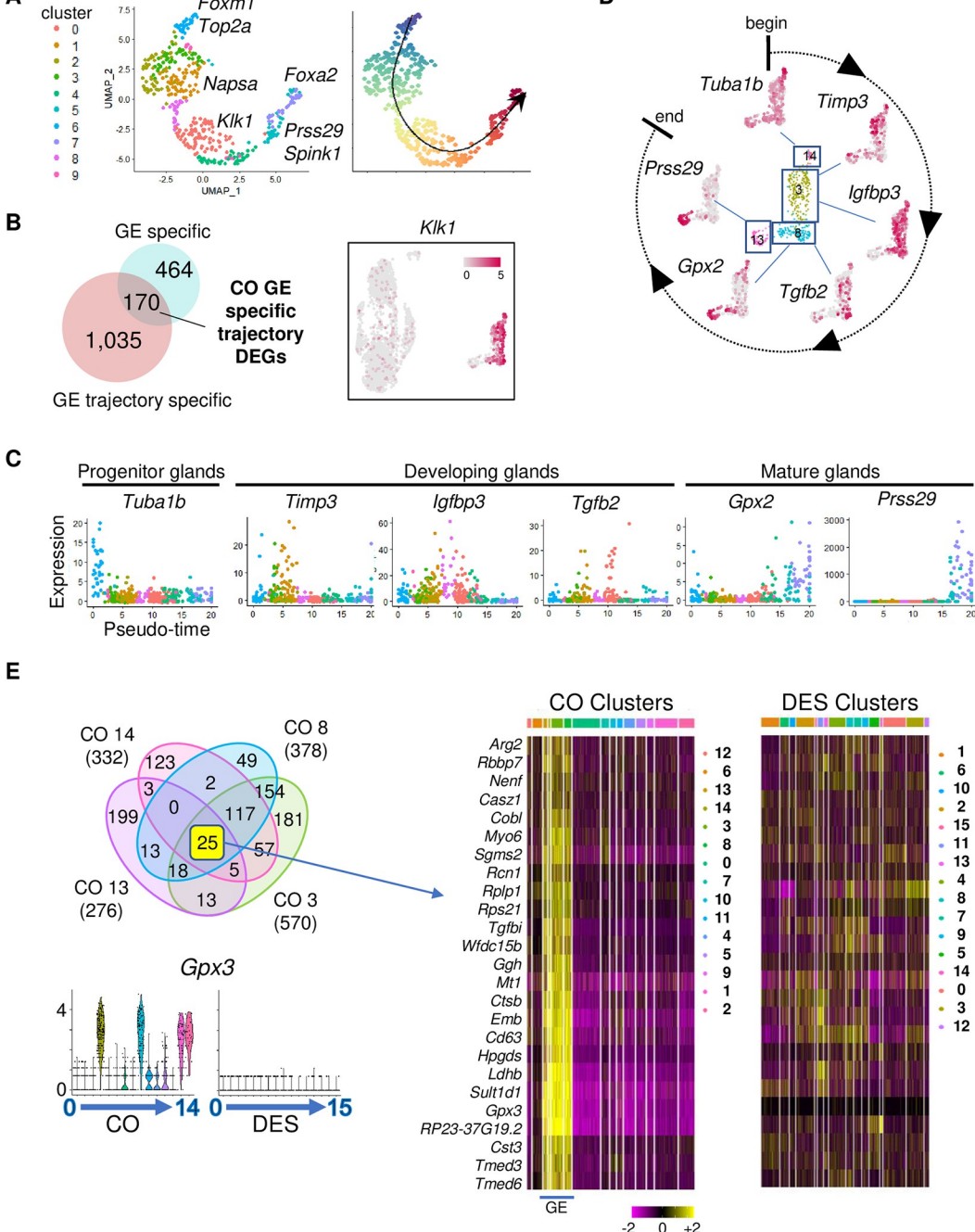

**Fig 4. GE differentiation trajectory determined by Slingshot analysis is unidentifiable in neonatal DES-exposed mice.**
(A) UMAP of Slingshot analysis of CO GE cells (left). Cluster numbers designated by color; select genes indicated.
Trajectory UMAP indicated by color (beginning, purple; end, red); arrow indicates direction of differentiation. (B) Venn
diagram of GE-specific DEGs overlapped with GE trajectory-specific DEGs. Feature plot of representative DEG from
overlapped gene list, *Klk1*; expression level is indicated. (C) Pseudo-time plots of select GE trajectory-specific DEGs. Each
dot represents 1 cell colored by cluster from panel A. (D) Feature plots of CO GE using select genes from panel C;
differentiation direction indicated. (E) Venn diagram of DEGs from CO GE clusters 3, 8, 13, and 14 (top left). Violin plots of
*Gpx3*, selected from 25 common DEGs (bottom left). Expression is natural log transformed counts. Hierarchical clustering
heat map of 25 common DEGs in CO GE cells; cluster number indicated by color at top. DES clusters plotted using the same
DEG order (right). Expression is centered (mean = 0 ± SD of each feature). The data underlying this figure can be found in
the Gene Expression Omnibus database under accession code GSE218156. CO, control; DEG, differentially expressed gene;
DES, diethylstilbestrol; GE, glandular epithelium; SD, standard deviation; UMAP, uniform manifold approximation and
projection.

*Foxa2* has long been used as a marker of GE; however, our scRNAseq data showed that *Foxa2* was expressed only in mature GE and in a small population of LE cells, findings that were validated at the protein level (Fig 3E). This observation is consistent with previously published localization of FOXA2 in adult uterus but differs from the ubiquitous presence of FOXA2 in the mature uterine glands of pregnant or pseudopregnant mice [41,42]. To find markers that would identify all developmental stages of GE but not LE, we selected the increased DEGs from all the CO GE clusters and overlapped them with each other (Fig 4E). Twenty-five DEGs were GE specific and found in all 4 CO GE clusters; a heat map confirmed their high exclusive expression in these clusters. For comparison, a heat map of the DES clusters for these 25 genes showed only sporadic expression in most clusters. Violin plots of one of these genes, *Gpx3*, showed expression restricted to the 4 CO GE clusters but no expression in any of the DES clusters, providing further evidence that DES-exposed mice lack normal GE. These findings provide a panel of markers that can be used for the identification of uterine GE cells at all stages of differentiation.

## Differentiation trajectory of luminal epithelium in controls

The large group of LE cells containing 10 clusters suggested that there were subpopulations of LE cells. Re-clustering the 10 CO LE cell clusters (without the GE clusters) resulted in 13 distinct clusters (Fig 5A and S8A Table). We used feature plots of *Top2a* and *Mki67* to identify cluster 11 as luminal progenitor cells (LPCs) (Fig 5B). Using this cluster as a starting point, a Slingshot analysis identified 7 trajectories that ended on 6 different clusters (Fig 5C). GO analysis of these 6 clusters revealed enrichment for distinct cellular functions including vascular morphogenesis (LE clusters 1 and 12), energy and metabolism (LE cluster 3), viral defense (LE cluster 7), inflammatory response (LE cluster 2), and extracellular matrix (LE cluster 10) (S8B Table). These data suggest that mature LE cells have distinct functions, despite their similar appearance.

## Disruption of uterine epithelial stem/progenitor cells in DES-exposed uteri

To identify EpSCs in CO epithelium clusters, we used markers previously reported as expressed in uterine stem cells (*Aldh1a1*, *Lgr5*, *Axin2*) [17,20,43,44]. Dual feature plots revealed high expression of all 3 markers in CO cluster 12, with some cells expressing combinations of these genes (Fig 6A). *Aldh1a1* and *Axin2* were also expressed in mature GE, while *Lgr5* and to a limited extent *Axin2* were also expressed in developing GE. These observations indicate that these particular markers cannot be used individually to identify stem cells; however, the expression of all 3 in the absence of *Foxa2* appears to identify the EpSC population. Because the uterine EpSC population has a very slow turnover rate [18], we used feature plots to identify clusters with high expression of cell division markers *Top2a* and *Mki67*. Only a few cells in the putative EpSC population had any *Top2a* or *Mki67* expression (Fig 6A). However, both markers were highly expressed in the GPC (CO cluster 14) and in the LPC (CO cluster 4; same cluster as CO cluster 11 in Fig 5B), which almost entirely lacked *Aldh1a1* expression (Fig 6A and S5A Table). An overlap of the 596 GPC up-regulated DEGs with the 502 LPC up-regulated DEGs showed 225 in common, suggesting they are related but distinctly different cell populations (S4A Fig and S9A–S9C Table). We interpret these results to indicate that CO cluster 12 consists of EpSC and the genes identified in S6C Table are candidates for EpSC-specific markers (Fig 3F).

To further investigate the connection between the CO EpSC, LPC, and GPC populations, we re-clustered these 3 populations and performed a Slingshot analysis. Choosing the EpSC population as the starting cluster, there were 2 trajectories that led separately to each of the PC

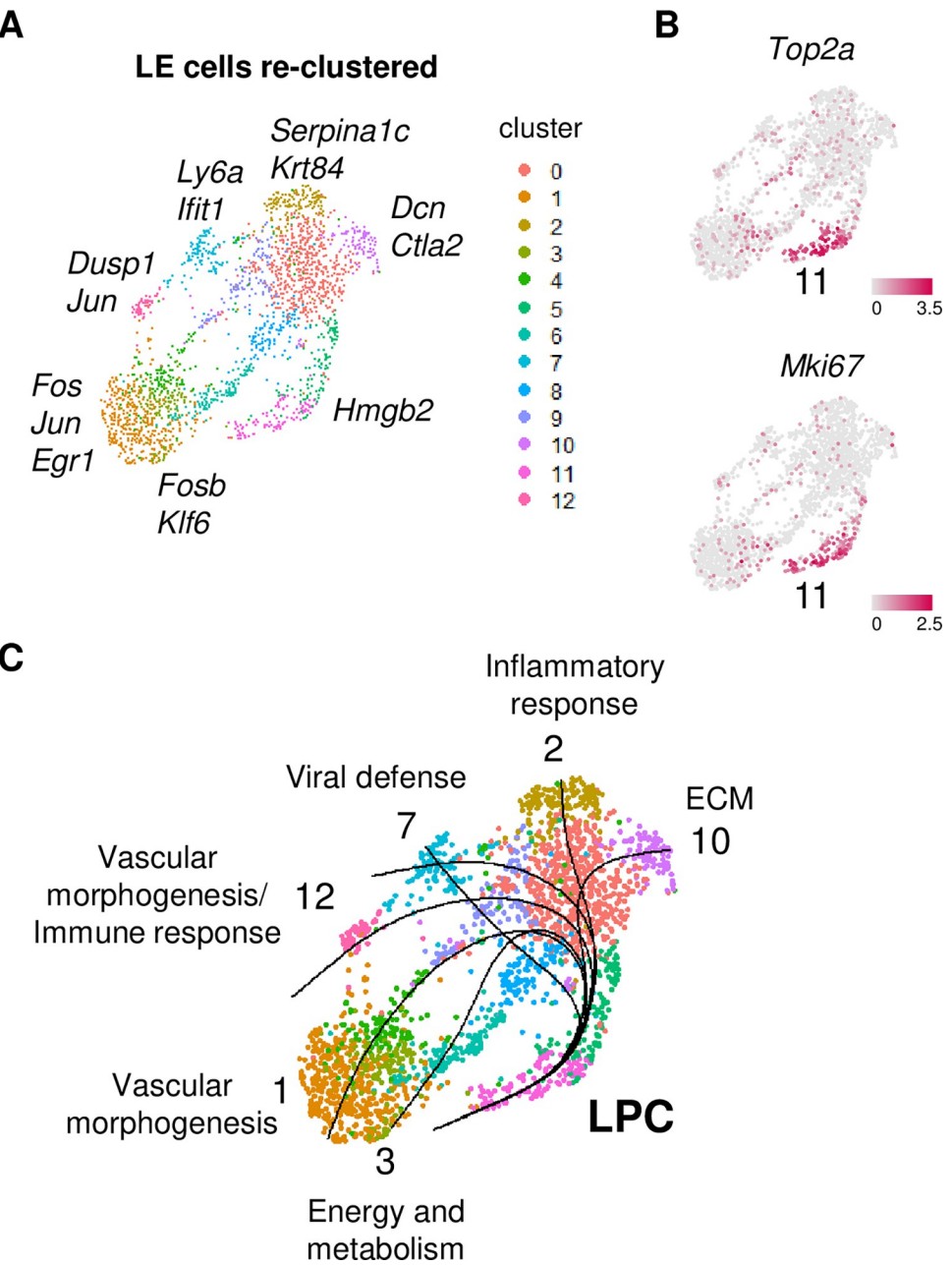

**Fig 5. LE differentiation trajectory determined by Slingshot analysis.** (A) UMAP of Slingshot analysis of CO LE cells. Cluster numbers designated by color; select highest DEGs in terminal clusters indicated. (B) Feature plots of *Top2a* and *Mki67* in the LE clusters. (C) Trajectory analysis indicating 7 paths of differentiation (black lines) ending on 6 different terminal clusters. The LPC cluster and the predominant highly enriched GO categories in each terminal cluster are indicated. The data underlying this figure can be found in the Gene Expression Omnibus database under accession code GSE218156. CO, control; DEG, differentially expressed gene; ECM, extracellular matrix; GO, Gene Ontology; LE, luminal epithelium; LPC, luminal progenitor cell; UMAP, uniform manifold approximation and projection.

populations (S4B Fig). To determine genes likely important for LPC or GPC differentiation, DEGs were determined for each of those lineages (339 DEGs EpSC+LPC lineage, CO12+CO4 versus CO14; 136 DEGs EpSC+GPC lineage, CO12+CO14 versus CO4) (S10A and S10B

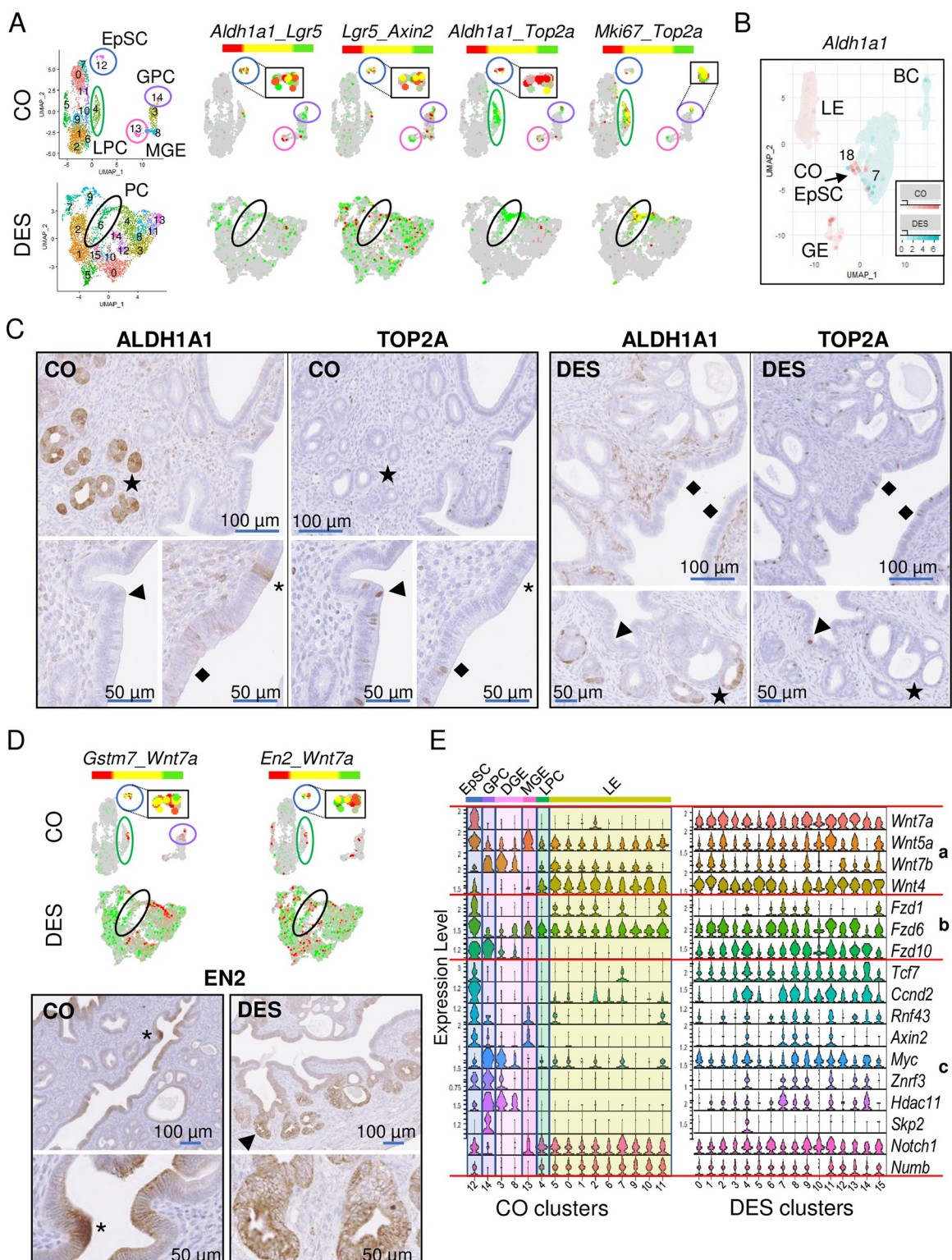

**Fig 6. EpSC and PC populations are substantially altered following neonatal DES exposure.** (A) UMAPs of non-integrated CO and DES epithelial cells. Epithelial cell populations identified as EpSCs (blue circle), GPCs (purple oval), LPC (green oval), MGE (pink circle), and PCs (black oval). Dual feature plots of select stem cell (*Aldh1a1*, *Lgr5*, and *Axin2*) and actively dividing cell (*Top2a* and *Mki67*) markers for CO (top) and DES (bottom). Insets in black boxes are EpSC or GPC magnified for clarity. (B) Dual feature plot of *Aldh1a1* in CO and DES epithelial cells from integrated UMAP (CO, pink; DES, teal). CO EpSC, GE, LE, and BC populations indicated.

(C) Representative ALDH1A1 and TOP2A IHC in adjacent sections from CO and DES uteri, as indicated. Stars (GE) or asterisks (LE) indicate cells that are ALDH1A1+/TOP2A-. Arrowheads (GPC or LPC) and diamonds (LPC) indicate cells that are ALDH1A1-/TOP2A+. Symbols correspond to same location in adjacent section; *n* = 4–6 mice per group. (D) Dual feature plots of *Gstm7*, *En2*, and *Wnt7a* for CO and DES. CO EpSC, GPC, and LPC populations indicated as in panel A. Insets in black boxes are EpSC magnified for clarity. Representative EN2 IHC in CO (bottom left) and DES (bottom right) uteri (*n* = 4–6 mice per group). Asterisk, EN2 at invaginating GE; higher magnification of same region below. Arrowhead indicates region of high EN2 in disorganized GE; higher magnification of same region below. (E) Violin plots of select Wnt ligands (a), receptors (b) and targets (c) in CO (left) and DES (right) epithelial clusters. Cluster numbers on x-axes; epithelial subtype in CO indicated at top. Expression is natural log transformed counts (y-axes). The data underlying this figure can be found in the Gene Expression Omnibus database under accession code GSE218156. BC, basal cell; CO, control; DES, diethylstilbestrol; EpSC, epithelial stem cell; GE, glandular epithelium; GPC, glandular progenitor cell; IHC, immunohistochemistry; LE, luminal epithelium; LPC, luminal progenitor cell; MGE, mature glandular epithelium; UMAP, uniform manifold approximation and projection.

Table). Violin plots of *Gstm7*, *Marcksl1*, and *Ptn* in the CO epithelial cell clusters from Fig 3B support these trajectories, with highest expression in the EpSC population, moderate expression in the relevant PC population, and then lower expression in the remaining cell clusters. However, only a small number of EpSC (71 cells) were captured in this analysis, so additional studies that include higher numbers of EpSC will be required for confirmation.

To determine if the DES epithelial clusters exhibited normal EpSC, GPC, or LPC expression patterns in specific clusters, dual feature plots of *Aldh1a1*, *Lgr5*, *Axin2*, *Top2a*, and *Mki67* were generated (Fig 6A). *Aldh1a1* was expressed in rare cells in DES cluster 4; however, it was not differentially expressed in any DES cluster (S5B Table). In addition, *Aldh1a1* expression was not overlapped by *Lgr5* or *Top2a*, suggesting these *Aldh1a1*+ cells were not EpSC or PCs. *Lgr5* and *Axin2* were present in most DES epithelial cell clusters with neither being DES cluster DEGs (Fig 6A and S5B Table). Both cell proliferation markers *Top2a* and *Mki67* were highly expressed only in DES cluster 6 (Fig 6A and S5B Table), suggesting that there was only 1 PC population in the DES epithelial clusters.

Because we failed to identify EpSC in the DES clusters using the published EpSC markers, we returned to the integrated (CO + DES) epithelial cell analysis (Fig 3A) to identify the DES clusters mapping most closely to CO EpSC. The CO EpSC in the integrated UMAP were identified by their expression of *Aldh1a1* in an integrated feature plot, with strong expression in only 2 regions: in a few GE cells and in a small set of cells adjacent to the large non-basal cell DES clusters (Fig 6B). The CO EpSC were located within integrated clusters 7 and 18, which were mainly comprised of DES cells, but most DES cells in these clusters expressed *Aldh1a1* at a low level or not at all (Fig 6B). The UMAP locations of the large set of non-basal DES epithelial cells near CO EpSC and far from the CO LE and CO GE indicates that even though DES EpSCs could not be identified, the non-basal DES epithelial cells were more closely related to CO EpSC than to LE or GE cells.

To test whether our identification of the stem and progenitor cell clusters was consistent with their spatial localization, we localized ALDH1A1 and TOP2A in uterine tissue. In CO uteri, ALDH1A1 was highly expressed in glands farthest away from the LE, consistent with its expression in mature GE and confirmed by the location of FOXA2 immunostaining in adjacent sections (Figs 6C and S5) [17,20]. ALDH1A1 was also expressed sporadically throughout the CO LE, consistent with previous reports of the EpSCs residing in this location [18]. In DES uteri, ALDH1A1 was occasionally expressed in the deepest GE, similar to controls; however, the vast majority of GE had no detectable ALDH1A1 (Fig 6C). In contrast to CO, there was no ALDH1A1 detected in the LE of the DES-exposed uteri. These data suggest that the *Aldh1a1* expression observed in non-integrated DES cluster 4 was from the GE cells. In CO, TOP2A was expressed sporadically across the LE as well as in invaginating GE, the region previously reported to contain epithelial stem/progenitor cells, supporting the scRNAseq identification of 2 distinct populations of CO PCs, GPC, and LPC (Fig 6C) [18,19]. In agreement with the

scRNAseq cluster analysis, ALDH1A1 and TOP2A protein expression was completely non-overlapping (Fig 6C). In DES uteri, TOP2A was expressed in many LE and GE cells. The presence of *Top2a+/Mki67+* cells only in DES cluster 6 suggests that these cells lack GE or LE identity, despite their localization in both epithelial regions in uterine tissue. ALDH1A1 and TOP2A immunostaining in serial sections of DES-exposed uteri clearly demonstrated that these 2 proteins did not overlap (Fig 6C), confirming the lack of *Aldh1a1* in the DES PC population.

We next attempted to identify EpSCs in DES clusters using genes highly expressed in the CO EpSC cluster. Two of the highest DEGs in this cluster, *Wnt7a* and engrailed 2 (*En2*), encode proteins in the Wnt/β-catenin signaling pathway (S5A and S6C Tables). *Wnt7a* is a secreted signaling molecule that regulates female reproductive tract differentiation including GE development, serving as a cell death suppressor in the uterus [45,46]. *En2* is a homeobox transcription factor that can repress *Wnt* signaling but is also an oncogene [47,48]. The top DEG in CO EpSC was *Gstm7*, and 3 additional *Gstm* genes were DEGs in this cluster (S6C Table). These genes encode mu family glutathione-S-transferases, which function to detoxify electrophilic molecules including products of oxidative stress [49,50]. Dual feature plots of *En2*, *Wnt7a*, and *Gstm7* in CO showed high expression of all 3 in EpSCs and sporadic expression in LPC and GPC clusters (Fig 6D). In contrast, all 3 genes were expressed in most DES clusters including DES cluster 6, the PC population, though they mainly were not expressed in the same cells (Fig 6D). In CO uteri, EN2 protein was quite restricted, with highest expression at the edge of budding GE and some expression in LE near these regions (Fig 6D). There was very low expression of EN2 in the CO GE. In DES uteri, EN2 was expressed in most epithelial cells including LE and GE, with some GE expressing very high levels. These data demonstrated a restricted pattern of stem/progenitor cell gene expression in normal uterine epithelial cells and the loss of this restriction in DES uteri, resulting in most epithelial cells expressing several stem/progenitor cell-specific genes.

The excessive aberrant expression of *Wnt7a* and Wnt signaling targets *Axin2* and *En2* in the DES cells suggested additional members of this pathway could be similarly disrupted. Violin plots of select Wnt signaling pathway ligands, receptors, and targets demonstrated moderate or high expression of most these genes in the CO EpSC population (Fig 6E). Other CO cell types had very restricted expression patterns. For example, *Wnt4*, *Fzd1*, and *Numb* were restricted to LE cell populations and *Wnt7b*, *Fzd10*, *Myc*, and *Hdac11* were restricted to developing GE. These data demonstrated tight regulation of Wnt ligands/receptors and their downstream targets in a cell type-specific manner. In DES clusters, there was widespread expression of most Wnt ligands, receptors, and targets (Fig 6E). These data show that DES epithelial cells exhibit severe disruption of the normally restricted expression pattern of Wnt signaling pathway genes and suggest widespread activation of Wnt signaling.

## DES-induced uterine cancer is characterized by activation of PI3K/AKT signaling

To identify cancer cells in the non-basal cell DES clusters, we used expression of *Six1*, a reliable biomarker of neonatal DES exposure-induced cancer [15,25,29,31]. *Six1* was significantly increased in DES clusters 5 and 14 with the highest fold change in DES cluster 5 (S5B Table). The cancer-associated genes *Olfm4* and *Rad51b* were among the top DEGs in DES cluster 5. *Olfm4* is both an oncogene and a target of Wnt signaling that provides negative feedback to this pathway, and *Rad51b* is a marker of DNA damage [51–53]. There was minimal sporadic expression of these markers in CO epithelial cells (Fig 7A). DES cluster 5 epithelial cells had high overlapping expression of *Six1* and *Olfm4* and almost exclusive expression and extensive overlap of *Olfm4* and *Rad51b* (Fig 7A). A feature plot of *Olfm4* in the spatial transcriptomics

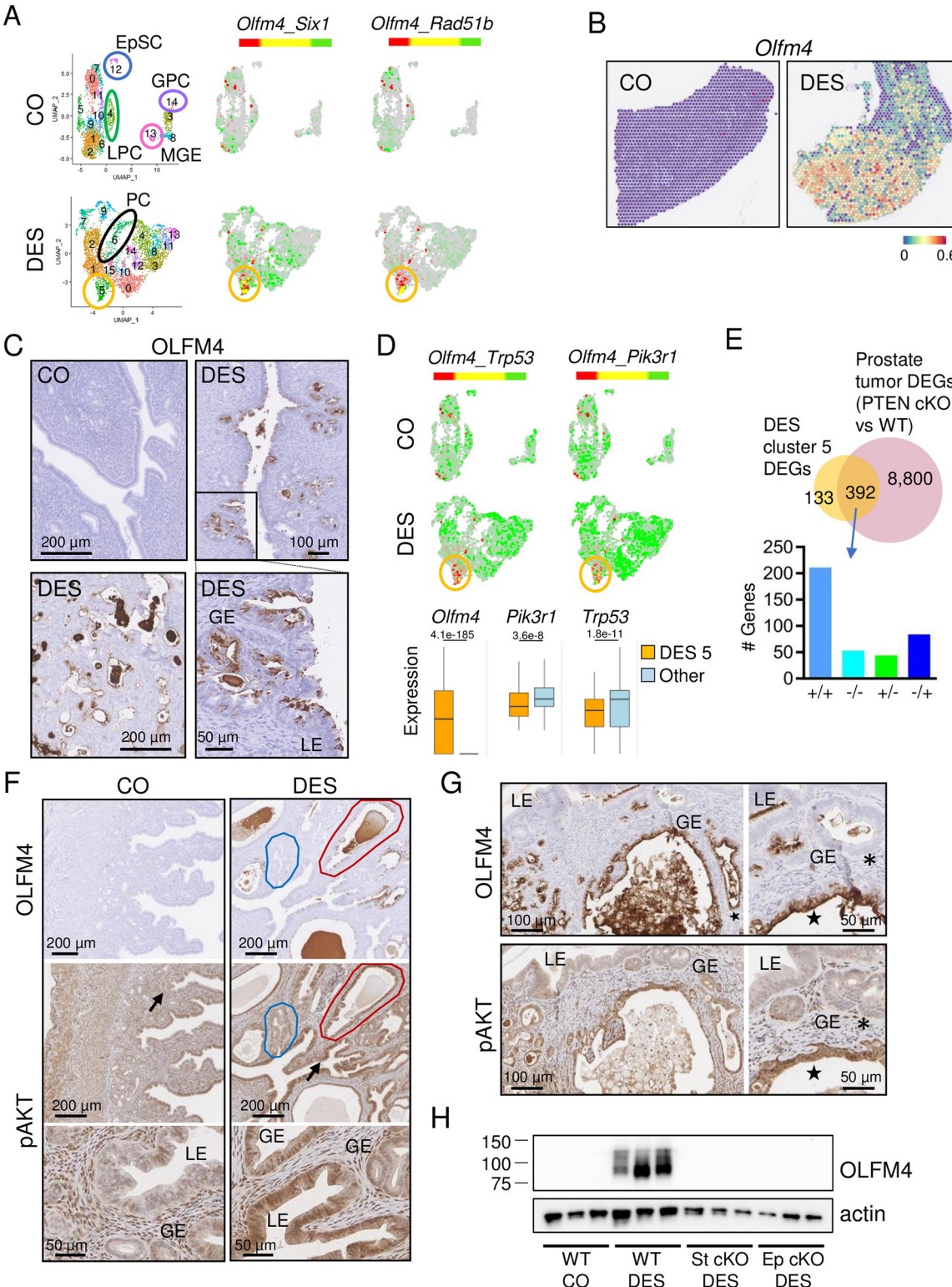

**Fig 7. Cancer cells in DES uteri are characterized by OLFM4 expression and PI3K/AKT signaling.** (A) UMAPs of non-integrated CO and DES epithelial cells. EpSC (blue circle), GPC (purple oval), LPC (green oval), and MGE (pink circle) are indicated in CO (top). PC (black oval) and cancer population (gold circle) are indicated in DES (bottom). Dual feature plots of cancer cell markers (*Olfm4*, *Six1*, and *Rad51b*) for CO (top) and DES (bottom). (B) Expression of *Olfm4* in spatial transcriptomics sections; CO (left), DES (right). Expression levels are natural log transformed counts. (C) Representative OLFM4 IHC in CO and DES sections (*n* = 4–6 mice per

group), as indicated. Bottom right section is magnified view of boxed region in top right section. Panels C, F, and G: LE, luminal epithelium; GE, glandular epithelium. (D) Dual feature plots of select cancer cell markers in CO (top) and DES (bottom). Cancer population, gold circle. Box and whisker plots of cancer markers for DES cluster 5 versus all other DES clusters. Expression levels are natural log transformed counts; adj *p*-values from S5B Table are indicated for each gene. (E) Venn diagram of DES cluster 5 DEGs and prostate tumor DEGs [prostate PTEN conditional KO (cKO) versus wild type (WT)]. Data generated from BaseSpace Correlation Engine knockout atlas (Illumina); original data from [54]. Overlapping genes categorized as increased (+) or decreased (-) in each study and split into 4 categories (+/+, -/-, +/- or -/+; DES cluster 5 DEGs listed first, prostate tumors listed second) on graph. (F) Representative OLFM4 and pAKT IHC in adjacent uterine tissue sections from 12-month-old CO (left) and DES (right) (*n* = 4–6 mice per group). Blue outline, region of low OLFM4/pAKT; red outline, high OLFM4/pAKT. Arrows indicate pAKT stained regions shown at higher magnification below. (G) Representative OLFM4 (top) and pAKT (bottom) IHC in adjacent uterine tissue sections from 9-month-old FVBN/J DES-exposed mice (*n* = 4–6 mice per group). Higher magnifications from same sections (right) (*n* = 4–6 mice per group). Asterisks, OLFM4-/pAKT- cells; stars, OLFM4+/pAKT+ cells. (H) Immunoblot of OLFM4 in uteri from 12-month-old control or DES-treated mice. Each lane has 5 μg uterine protein extract from 1 mouse (*n* = 3 mice per group). Actin used as a loading control. WT, wild type; St cKO, stromal ERα conditional knockout; Ep cKO, epithelial ERα conditional knockout. Raw immunoblots in S1 Raw Images. The data underlying this figure can be found in the Gene Expression Omnibus database under accession code GSE218156. CO, control; DEG, differentially expressed gene; DES, diethylstilbestrol; EpSC, epithelial stem cell; GPC, glandular progenitor cell; IHC, immunohistochemistry; LPC, luminal progenitor cell; MGE, mature glandular epithelium; UMAP, uniform manifold approximation and projection.

uterine tissue sections showed high expression in cancer regions in the DES sample and a lack of expression in the CO (Fig 7B). To validate DES cluster 5 as the cancer cell population, we performed OLFM4 immunohistochemistry (IHC). OLFM4 was not detected in CO uterine tissue but was robust in LE and GE cells in cancer regions in DES-exposed uterine tissue (Figs 7C and S6A). In addition, OLFM4 was expressed in cells inside gland lumens. These cells are commonly observed in cancer regions in DES-exposed mice, but their cell type is not yet determined. OLFM4 expression in LE and GE was variable across animals, with a direct correlation between extent of uterine cancer and extent of OLFM4 staining (Figs 7C and S6A). These data confirmed that DES cluster 5 was the uterine cancer cell population and identified *Olfm4* and *Rad51b* as additional markers of this cancer type.

To identify drivers of the cancer phenotype, we next examined DEGs in the DES cancer cell cluster. Activating mutations in β-catenin, which are accompanied by activation of Wnt signaling, are associated with human endometrioid endometrial carcinomas [55]. This observation, combined with our previous data showing widespread activation of Wnt signaling in DES epithelial cells, suggested that Wnt signaling was driving the DES cancer phenotype. Examination of DES cluster 5 for expression of Wnt signaling targets, however, indicated that relative to other DES clusters, the DES cancer cells had among the lowest expression of Wnt ligands, receptors, and target genes including *Wnt7a*, *Ccnd2*, *Axin2*, and *Myc* (Fig 6E). These findings were inconsistent with Wnt signaling as the sole cancer driver.

In the mouse, loss of the tumor suppressor PTEN in the uterus leads to rapid development of endometrial cancer, even if it is only deleted in the epithelial cells, and combined loss of PTEN and a second tumor suppressor, TRP53, induces higher grade cancer with invasion [56,57]. *Trp53* and *Pik3r1*, which encodes a tumor suppressor that stabilizes PTEN and restrains PI3K catalytic activity [58,59], were widely expressed in most CO and DES clusters but were down-regulated in DES cluster 5 (Fig 7D and S5B Table). Dual feature plots showed an inverse relationship between *Olfm4* and both *Trp53* and *Pik3r1*; the differences between expression of these 3 genes in DES cluster 5 relative to all other DES clusters were highly significant (Fig 7D).

The reduction in expression of *Pik3r1* in the cancer cell population suggested a role for increased PI3K/AKT signaling in DES cancer formation. To further examine the molecular signature of the DES cancer cells, we compared the DES cluster 5 DEGs to curated gene perturbation models in the BaseSpace Correlation Engine knockout atlas (Illumina). The gene perturbation with the highest correlation to this dataset was PTEN, which blocks PI3K activity. One dataset with high overlap was a conditional deletion of *Pten* in prostate epithelium, which

results in prostate tumors [54] (Fig 7E and S11 Table). Of the DES cluster 5 DEGs, 392/525 (75%) overlapped the DEGs in this prostate cancer model; 67% of these genes were altered in the same direction (Fig 7E). These data suggest that increased PI3K activity plays a major role in the formation of neonatal DES exposure-induced uterine cancer.

To test whether PI3K/AKT signaling was activated in DES-induced uterine cancer cells, we localized OLFM4 and phosphorylated AKT (pAKT) in adjacent sections. CO uteri (during estrus) had diffuse pAKT immunoreactivity in most cell types but only sporadically in the nuclei of LE cells and lower staining in GE compared to stroma (Fig 7F). DES uteri had higher pAKT nuclear and cytoplasmic immunoreactivity in LE cells compared to CO and there was very strong pAKT nuclear and cytoplasmic staining in most GE. The regions of more intense pAKT staining generally corresponded with regions of OLFM4 expression (Fig 7F). Because DES-exposed CD-1 mice only develop localized foci of uterine adenocarcinoma by 12 months of age, we further explored the correlation of OLFM4 and pAKT in a more robust model of developmental DES-induced uterine cancer. Female FVBN/J mice develop uterine adenocarcinoma at an earlier age (6 months) and they exhibit very extensive adenocarcinoma throughout the uterus by 9 to 12 months of age following the same neonatal DES exposure paradigm used in our standard CD-1 mouse model [31]. Uterine tissue sections from 9-month-old DES-exposed FVBN/J mice were immunostained for OLFM4 and pAKT in adjacent sections. As previously observed, FVBN/J mice exhibited more extensive cancer regions. These regions had more robust OLFM4 expression relative to that in CD-1 mice (Figs 7F and 7G and S6A and S6B). Staining of adjacent sections for pAKT revealed consistent overlap of pAKT with OLFM4+ cancer regions (Figs 7G and S6B).

The data presented so far suggest a model in which abnormally differentiated epithelial cells are influenced by stromal inflammation and oxidative stress (Fig 2F) to activate PI3K/AKT signaling that then drives uterine adenocarcinoma development. However, we have not yet demonstrated that DES-induced changes in epithelial or stromal gene expression are essential for the cancer phenotype. To answer this question, we generated conditional knockout (cKO) mice lacking ERα only in endometrial stromal cells or only in endometrial epithelial cells [60,61]. The loss of ERα in these tissue compartments precludes DES action on the targeted cell type. Uteri were collected from 12-month-old control mice and neonatal DES-exposed wild type, stromal ERα cKO, and epithelial ERα cKO mice. Unfortunately, although control uteri could be evaluated accurately, the very thin nature of uterine tissues from both cKO lines resulted in inadequate sections through the lumen of several samples, precluding a thorough histologic assessment of cancer incidence. Uterine tissue sections from 3 to 4 mice per group, however, confirmed the presence of cancer in DES-exposed controls and the lack of cancer in DES-exposed stromal ERα cKO and epithelial ERα cKO groups (S7A–S7C Fig). DES-exposed stromal ERα cKO mice had a columnar LE with some invaginations of the uterine lumen and limited GE development. DES-exposed epithelial ERα cKO mice had a simple tubular lumen lined with cuboidal LE but no apparent GE. Based on our identification of OLFM4 as a cancer marker, we then tested for the presence of uterine cancer by immunoblotting. DES-exposed wild-type mice had high levels of OLFM4, but neither controls nor either of the DES-exposed cKO lines expressed this cancer marker (Fig 7H). These findings indicate that ERα-dependent DES-induced changes in cellular differentiation pathways of both epithelial and stromal cells are required for the development of uterine adenocarcinoma.

## Discussion

The dataset reported here provides a rich resource of scRNAseq information from over 48,000 total adult mouse uterine cells, allowing an in-depth analysis of cell types. Previous scRNAseq

analyses of rodent uterine cells reveal interesting information regarding postnatal uterine development [17,22,23] and pregnancy [62,63]. Two datasets contain cells from adult non-pregnant uterus, but both are limited by having few cells for analysis [17,64]. Here, we focused mainly on epithelial cells because they are the cell type that develops into uterine adenocarcinoma following a developmental insult. The large number of epithelial cells from adult control females in estrus enabled clear identification of stem cells, progenitor cells, luminal cells, and glandular cells, including luminal and glandular cell developmental trajectories.

Our findings confirm previous observations that uterine EpSC are localized at the intersection between glands and lumen [18]. However, instead of EpSCs differentiating into a single PC population [18], we show that they differentiate into 2 distinct proliferating PC types, one destined to become luminal epithelium and one to become glandular epithelium (Fig 8). The EpSCs are marked by high expression of EN2 protein and *Wnt7a*, but neither marker is entirely specific to stem cells, and the presence of ALDH1A1 in mature glands argues against its previously suggested specificity as a marker of stem/progenitor cells [17]. Excitingly, we identified a new uterine EpSC marker, *Gstm7*, and found that several additional *Gstm* isoforms were highly expressed in EpSC. This finding mirrors previous observations of increased *GSTM* family gene expression in human fetal liver hematopoietic stem cells, including *GSTM2*, the human *Gstm7* homolog [65]. Enrichment of enzymes that function in a pathway responsible for managing and eliminating reactive oxygen species from the cell makes sense for uterine EpSCs, which must be preserved for the lifetime of the individual. These new progenitor and stem cell markers, however, must be validated by future functional studies such as lineage tracing experiments.

DES exposure permanently disrupts differentiation of uterine epithelium. One of the most striking results of this altered differentiation is the appearance of basal cells in the uterus, a tissue that does not normally contain this cell type [15,25,29]. Interestingly, basal cells do not substantially contribute to the cancer phenotype because mouse uteri lacking the oncoprotein *Six1* do not develop the basal cell phenotype but still have cancer [29]. In the current study, we show that the epithelial cells are abnormal and have characteristics of luminal and glandular epithelium as well as EpSCs, but no distinct EpSC population. Instead, we find a single PC population that could not be further characterized as glandular or luminal, suggesting substantial problems with the downstream fate of these cells. The abnormal epithelial differentiation trajectory was confirmed by the constitutive activation of Wnt signaling in most DES epithelial cells, in stark contrast to the epithelial subtype specificity observed in differentiated CO epithelial cells. Increased Wnt signaling due to activating mutations in pathway members can cause human endometrial cancer [66,67]. However, it is unlikely that Wnt signaling alone explains neonatal DES-induced uterine cancer because persistent activation of Wnt signaling in the mouse uterus using genetic approaches results in uterine hyperplasia but not cancer [68,69]. In addition, the DES cancer cell cluster had reduced Wnt pathway activation relative to the other DES epithelial cell clusters. Instead, this cluster had diminished *Trp53* and *Pik3r1* mRNAs and elevated levels of PI3K/AKT signaling, which are common drivers of many cancer types, including endometrial cancer in humans and animal models [56,66,67,69,70]. The impact of abnormal stromal cell inflammatory and oxidative stress responses on the adjacent epithelium cannot be ignored in this model because both DES-initiated stromal and epithelial changes are required for cancer development. Overall, these findings provide compelling evidence that abnormal activation of PI3K/AKT signaling is the long-sought explanation for uterine cancer development following developmental estrogen exposure (Fig 8).

The DES model has a striking resemblance to a mouse model of intestinal tumorigenesis induced by expression of a constitutively active form of β-catenin in intestinal epithelial cells (IECs) [71]. In the IEC model, active β-catenin induces dedifferentiation of epithelial cells,

## Normal Differentiation

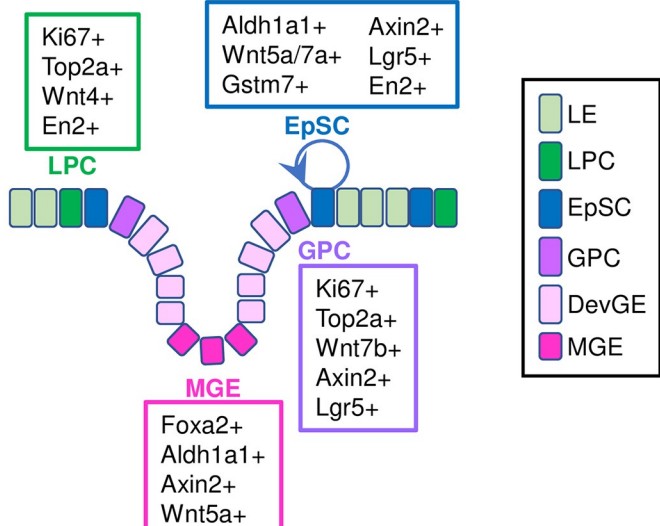

## Neonatal DES Exposure

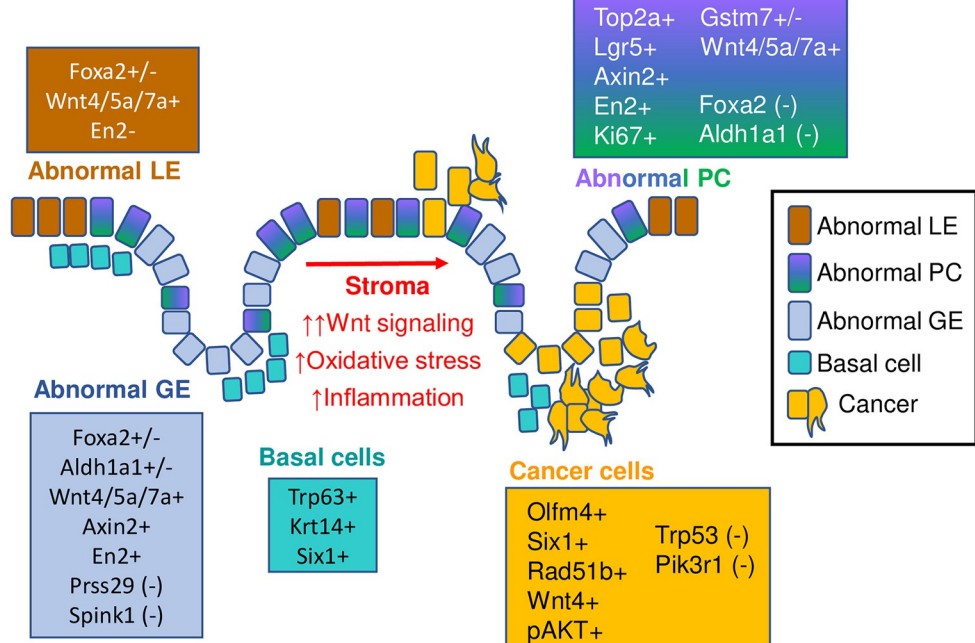

**Fig 8. Model of uterine epithelial cell differentiation in control and DES-exposed uteri. Normal differentiation:** Rare EpSCs transition to proliferative LPCs or GPCs. LPC transition to LE and GPC transition to DevGE and then MGE. **Neonatal DES exposure:** Proliferative abnormal PCs transition to either abnormal LE or abnormal GE, which are distinguished mainly by their histological locations. Basal cells form an extra layer under both LE and GE. Widespread activation of Wnt signaling in the presence of stromal oxidative stress and inflammation activate PI3K/ AKT signaling to drive malignant transformation of abnormal epithelial cells. Markers present at the various stages of differentiation for control and DES-exposed uteri are indicated. DES, diethylstilbestrol; DevGE, developing glandular epithelium; EpSC, epithelial stem cell; GE, glandular epithelium; GPC, glandular progenitor cell; LE, luminal epithelium; LPC, luminal progenitor cell; MGE, mature glandular epithelium; PC, progenitor cell.

which gain stem cell characteristics. β-catenin induces activation of the inflammatory mediator nuclear factor-κB (NF-κB), which serves to amplify β-catenin-mediated signaling and induction of IEC tumor formation. Two genetic mouse models that modify the levels of β-catenin in the uterus also recapitulate some of the aspects of DES-induced phenotypes: constitutively active β-catenin results in endometrial hyperplasia and loss of β-catenin results in squamous cell metaplasia (abnormal basal cell differentiation) [68]. Similarly, the DES model has incomplete differentiation of epithelial cells associated with widespread activation of Wnt signaling, combined with inflammation in the stroma as a cell proliferation signal. The main difference between the IEC and DES models is that the IEC model requires genetic manipulation to drive β-catenin signaling, whereas the DES model simply requires estrogen-mediated signaling for a brief period during epithelial cell differentiation. The similarity between these models highlights the influence of environmental exposures during development on tumor susceptibility later in life.

An important question remains: How does PI3K/AKT signaling become activated in the cancer cells? Because most of the epithelial cells, not just the cancer cells, have persistent activation of Wnt signaling, it is likely that this signaling pathway is activated because of the "first hit" of DES exposure and sets the stage for later activation of PI3K/AKT in the cells that become malignant. There are many connections between Wnt and PI3K/AKT signaling pathways, including common components relevant to cancer development such as MYC, GSK3, PTEN, and CCND1 (reviewed in [69]). Recently, WNT3A-mediated activation of PI3K was demonstrated in colorectal cancer cells [72]. Additional connections between these pathways could be mediated by noncoding RNAs such as microRNAs or long noncoding RNAs [73]. Finally, stromal inflammation- and oxidative stress-induced genotoxicity and cell proliferation may promote cancer development through activation of NF-κB and PI3K/AKT signaling in epithelial cells [74,75]. NF-κB appears to be activated in the DES cancer model based on pathway analysis of altered genes, including *Olfm4*, which can be up-regulated by NF-κB, among other transcription factors [52,76]. A bidirectional interaction between the inflammatory stroma and the Wnt-activated epithelial cells likely contributes to cancer development in this model.

The most frequently mutated genes in human endometrial endometrioid cancer include the tumor suppressors PTEN and PIK3R1; the overwhelming majority of these cancers have abnormal activation of the PI3K/AKT signaling pathway [58,66,67]. In human endometrial organoids, inhibition of Wnt signaling increases epithelial cell differentiation [77], suggesting that persistent Wnt signaling in human endometrium, as in the mouse, could reduce epithelial differentiation. Although most human endometrial cancers appear to result from a primary mutational event, perhaps some result from an early environmental insult during cellular differentiation, leading to Wnt pathway activation and abnormal differentiation of epithelial cells. Later induction of PI3K/AKT signaling could be related to stromal inflammation and subsequent induction of mutational events through increased oxidative stress, down-regulation of *PI3KR1* mRNA, or other alterations in regulators of the PI3K/AKT pathway.

## Methods

### Key resources

All resources used in the manuscript can be found in the Key Resources Table (S12 Table).

### Ethics statement

The National Institute of Environmental Health Sciences (NIEHS) animal care and use committee approved this research under protocol #2007–0038. Animals were maintained in

accordance with the National Research Council Guide for the Care and Use of Laboratory Animals, the Animal Welfare Act and Regulations, and the provisions of the National Institutes of Health Intramural Research Program Animal Welfare Assurance.

## Animals

CD-1 mice were obtained from the NIEHS in-house breeding colony and housed under conditions previously reported [33]. FVBN/J mice were purchased from The Jackson Laboratory (Jax#001800) to generate female pups for experiments described below. Previous work demonstrated that C57BL/6 mice do not develop uterine adenocarcinoma; therefore, we first generated mouse lines for ERα (*Esr1*) flox, *Amhr2*-cre, and *Wnt7a*-cre that were backcrossed to an FVBN/J background for 10 generations [60,61,78]. ERα (*Esr1*) cKO mice were generated by crossing *Esr1* floxed mice (FVBN/J background) with either *Amhr2*-cre mice (stromal cKO; FVBN/J background) or *Wnt7a*-cre mice (epithelial cKO; FVBN/J background); proper names are listed in the Key Resources Table (S12 Table).

## Animal treatments and tissue collection

Briefly, female pups delivered from timed pregnant dams listed above were randomly distributed to 10 female pups per litter and then randomly assigned treatment groups of either corn oil (CO) or DES (1 mg/kg/day) (Sigma-Aldrich). Pups were exposed on neonatal days 1 to 5 by subcutaneous injection using a volume of 0.02 mL; mice were weaned and housed as described previously [33]. CD-1 mice exposed to DES at this dose and timing develop uterine adenocarcinoma (incidence >50%) at 12 months of age, so we collected uteri for single-cell isolation, frozen tissue sections, and paraffin embedded sections at this age from 3 independent groups of mice [15]. Adult DES-exposed mice are in persistent estrus; therefore, CO mice in estrus were selected using vaginal observation [79,80]. Uteri from FVBN/J mice were collected at 9 months of age for IHC. For immunoblotting, uteri were collected at 12 months of age from control and DES-exposed wild type (*Esr1* flox/flox), epithelial cKO (*Esr1* flox/flox, *Wnt7a*-cre+), and stromal cKO (*Esr1* flox/flox, *Amhr2*-cre+) mice and frozen at −80°C until use.

## Single-cell isolation from adult uterine tissue

Mice were euthanized and uterine horns were excised away from the uterine body and the oviducts. Uteri from 2 CO and 4 DES mice were pooled for single-cell isolation. All procedures were performed on ice unless otherwise specified. Uteri were rinsed in calcium and magnesium free phosphate buffered saline (PBS-CMF); horns were slit open lengthwise and soaked in PBS to remove any debris. Cell dissociation was performed by incubating tissue in Trypsin-EDTA (0.25%) (Gibco-Thermo Fisher) on ice for 1 h, then 10 min at room temperature, then 50 min in a 37°C water bath with gentle agitation. Tissue was removed from media, and the cell suspension was centrifuged at $450 \times g$ for 10 min at 4°C. Pellets were resuspended in DMEM/F12 (Gibco-Thermo Fisher) containing 10% heat inactivated fetal bovine serum (FBS; Thermo Fisher), serially filtered through sterile CellTrics 100 μm and 30 μm filters (Fisher Scientific), and centrifuged at $450 \times g$ for 10 min at 4°C. Pellets were resuspended in 1 to 2 mL PBS-CMF containing 0.04% AlbuMAX bovine serum albumin (BSA; Thermo Fisher). A small aliquot of cell suspension was stained with Trypan Blue (Gibco-Thermo Fisher) and counted using a hemocytometer.

## Single-cell RNA sequencing

Single-cell libraries were prepared using the Chromium platform with the Chromium Single Cell 3′ Reagent Kit v3 (Cat. 1000075, 10x Genomics) following the manufacturer's protocol.

Briefly, freshly prepared single cells and single gel beads conjugated with cell barcodes and reverse transcription primers were partitioned into oil droplets as emulsion in the 10x Genomics Chromium Controller instrument followed by cell lysis and barcoded reverse transcription of mRNA, cDNA amplification by PCR, fragmentation, and adding adapters and sample index amplification by PCR. Libraries were sequenced on an Illumina NovaSeq 6000 for paired end reads: read 1, 30 bp; read 2, 100 bp.

The scRNA count matrix was generated using cellranger v3.0.1, using GENCODE genes for mm10 (downloaded from 10x Genomics' website on 3/22/2018), which were filtered according to 10x Genomics recommendations. Potential barcode swapping was identified using the R package DropletUtils v1.14.2, and potential cell doublets were identified and removed using the R package scran v1.22.2. Downstream analysis was carried out using the R package Seurat v3.1.0 following the standard SCTransform-based pipeline with MAST used for differential expression testing. The only exception to this methodology was for the analysis associated with Fig 1A, in which sample integration was carried out following Seurat's CCA integration vignette to more easily identify cell type by limiting the differences between CO and DES gene expression. The sequencing data were deposited in the Gene Expression Omnibus database under accession code GSE218156 and are publicly available.

## Spatial transcriptomics

The Visium Spatial Gene Expression Slide kit (10x Genomics, Cat. 1000184) was used for the Spatial Transcriptomics study. Uterine tissue freezing and embedding was performed following the 10x Genomics Visual Spatial Protocol-Tissue Preparation Guide, CG000240-Rev A. Briefly, uterine tissues were frozen in isopentane and embedded in chilled Tissue-Plus O.C.T. compound (Fisher Scientific) on dry ice. Tissue sections were stained with Mayer's hematoxylin (Millipore Sigma) and eosin (Millipore Sigma) (HE) and evaluated by a board-certified veterinary pathologist for the presence of luminal and glandular epithelial cells in both CO and DES sections and the presence of uterine adenocarcinoma in the DES sections ($n = 5$ mice per group). Two mice per group were selected for subsequent sectioning for spatial transcriptomics. Selected blocks were cored to capture the areas of interest and fit the capture area on the Gene Expression slide. Tissues were cryo-sectioned at 10 μm thickness and immediately placed on the Gene Expression slide. After 4 areas were captured (2 control and 2 DES), the slide was kept at −80°C until staining. The slide was fixed with methanol at −20°C for 30 min and stained with HE per manufacturer's instructions. The slide was imaged using an Aperio AT2 slide scanner (Leica Biosystems). Optimal permeabilization time of 18 min was determined using the tissue optimization kit (10x Genomics, Cat. 1000193). On-slide mRNA reverse transcription, cDNA synthesis and release, cDNA amplification, and library preparation were performed following instructions in the user manual. The libraries were then sequenced on a NextSeq 500 for paired end reads: read 1, 28 bp; read 2, 90 bp. The spatial sequencing data was processed using 10x Genomics Space Ranger v1.2.2 and analyzed using Seurat v4.0.5. The reference genome used for the scRNAseq analysis was mm10. For all differential gene expression tables, the percent expressing cells columns are the percentage of cells that are expressing (count >0) that gene in the cluster indicated and then the percentage of cells expressing that gene in all other clusters. The sequencing data were deposited in the Gene Expression Omnibus database under accession code GSE218156 and are publicly available.

## Immunohistochemistry

Whole uterine tissues from CO and DES mice ($n = 4$–6 mice per group) were fixed in 10% neutral buffered formalin for approximately 48 h, changed into 70% ethanol, processed and

embedded in paraffin blocks, and sectioned at 6 μm thickness. Briefly, slides were deparaffinized in xylene, rehydrated through graded ethanol, and blocked for endogenous peroxidases using 3% hydrogen peroxide for 15 min. IHC methods have been described in detail previously [81]. All concentrations and detailed protocol instructions for each antibody can be found in S13 Table. Protein/antibody complexes were visualized by using 3-diaminobenzidine chromagen (Dako) for 6 min, counterstained with hematoxylin, dehydrated and cover slipped. Slides were scanned using an Aperio AT2 slide scanner (Leica Biosystems). Images were captured using Aperio ImageScope v. 12.4.3.5008 (Leica Biosystems).

## Immunoblotting

Uterine tissues were pulverized on dry ice and total protein isolated using TPER (Thermo Fisher). Protein concentration was assessed using a Qubit protein assay (Thermo Fisher). Samples (5 μg) were loaded on a 10% TGX gel (Bio-Rad), run at 150 V, and transferred to PVDF membrane using the Trans-Blot Turbo transfer system (Bio-Rad). The blot was blocked in 5% Blotto (Thermo Fisher) in tris buffered saline plus 0.1% Tween-20 (TBS-T) for 1 h at RT. Rabbit monoclonal anti-OLFM4 (Cell Signaling Technology) was diluted to 0.6 μg/mL in 5% blocking solution and applied to the blot overnight at 4°C. The blot was washed 3 times for 15 min with TSB-T, and incubated with donkey anti-rabbit IgG diluted 1:25,000 in 1% blocking buffer for 1 h at RT. Following 3 washes in TBS-T for 15 min each, immunoreactive bands were visualized using Super Signal West Femto reagents (Thermo Fisher) following the manufacturer's instructions. Images were captured using a ChemiDoc Touch Gel Doc system (Bio-Rad). Antibodies were stripped with Restore (Thermo Fisher) for 30 min at 37°C. Peroxidase conjugated mouse monoclonal anti-β-actin (Sigma) at a dilution of 1:10,000 in 5% blocking solution was applied for 1 h at RT and visualized as described above.

## Quantification and statistical analysis

For immunohistochemical analysis, a minimum of 4 to 6 mice per group were immunostained with each antibody and representative images are shown. For immunoblotting, 3 mice per group were tested for OLFM4—all samples are shown in immunoblot in Fig 7H. For scRNA-seq, statistical analysis was carried out using the R package Seurat v3.1.0 following the standard SCTransform-based pipeline with MAST used for differential expression testing. For spatial transcriptomics, statistical analysis was carried out using 10X Genomics Space Ranger v1.2.2 and analyzed using Seurat v4.0.5. Statistical analysis for data presented as box and whisker plots in Fig 6C are from S5B Table.

## Supporting information

**S1 Fig. Identification of uterine cell types captured by single-cell RNA-seq.** (A) Heat map of mesothelial and stromal cell markers in CO and DES epithelial cells. Expression is Pearson Residuals from the SCTransform method. UMAP is from Fig 1A with cell types indicated by color. (B) Dual feature plots of epithelial and basal cell markers (*Krt18*, *Krt14*, *Six1*, and *Trp63*). CO (left) and DES (right). Colors for each gene (red or green) are indicated above the UMAPs and yellow indicates overlapping expression. (C) Feature plots of stromal markers *Col6a3*, *Dpt*, and *Vcan* and epithelial marker *Epcam* using the integrated UMAP of all cells from CO (peach) and DES (teal). Expression is the same as in A. The data underlying this figure can be found in the Gene Expression Omnibus database under accession code GSE218156. (TIF)

**S2 Fig. Stromal cell gene expression is altered by DES as evidenced by spatial transcriptomics.** (A) Uterine tissue sections from CO B and DES B used for spatial transcriptomics (ST). HE stain of tissue section adjacent to ST section (left) and tissue sections used for ST (Slide-seq HE, middle). Cluster identification using Space Ranger-1.2.2, skmeans 10 (10x Genomics); colors represent distinct clusters. (B) Heat map of select uterine tissue cell type markers plotted for CO B (left) and DES B (right). Cell type is indicated below heat maps (M = muscle, S = stroma, E = epithelium, BC = basal cells). Expression values are Pearson Residuals from the SCTransform method. (C) ST section of select markers for CO and DES; expression = natural log transformed counts. (D) Venn diagrams of stromal cell GO categories with ≥10 genes (compared groups indicated). (E) Representative FOXL2 IHC in CO and DES; $n$ = 4–6 mice per group. The data underlying this figure can be found in the Gene Expression Omnibus database under accession code GSE218156.
(TIF)

**S3 Fig. Epithelial cells from DES-exposed mice lack subtype identity.** (A) Heat map of top DEGs of each cluster from integrated epithelial cell UMAP. Expression is Pearson Residuals from the SCTransform method. Clusters are indicated across the top and select clusters indicated: Basal cells (8, 11); GE (13, 17, and 19); overlapping CO and DES clusters (7 and 18). (B) Violin plots of developing GE markers (*Prap1*, *Hif1a*, and *Wnt7b*) for CO and DES. Cluster numbers are indicated below violin plots. Expression is natural log transformed counts and is indicated for each gene. (C) Representative FOXA2 IHC in CO and DES ($n$ = 4–6 mice per group). LE and GE indicated. (D) Dot plots of 8 LE-specific genes in CO and DES epithelial clusters. Expression and gene count indicated. Feature plots of *Adgrg7* expression in non-integrated CO and DES epithelial clusters as well as integrated CO and DES epithelial clusters. Expression levels are indicated. (E) Violin plots of top 5 CO12 and LE markers from Fig 3F in DES clusters. Cluster numbers are indicated below violin plots. The data underlying this figure can be found in the Gene Expression Omnibus database under accession code GSE218156.
(TIF)

**S4 Fig. EpSC, LPC, and GPC trajectory analysis.** (A) Overlap of up-regulated LPC DEGs (CO cluster 4) with GPC DEGs (CO cluster 14). Ten highest fold-change DEGs in each category indicated. (B) Re-clustering of EpSC, LPC, and GPC with Slingshot analysis. Trajectory analysis using EpSC as the starting point showed 2 paths of differentiation (black lines). Top DEGs for each trajectory grouping (EpSC+GPC vs. LPC or EpSC+LPC vs. GPC); gene expression was filtered by the percent of cells expressing each DEG in each grouping compared to the remaining cluster; numbers in parentheses indicate the percent cells expressing in each group. Violin plots of CO epithelial clusters from Fig 3B of EpSC+GPC lineage genes, *Gstm7* and *Marcksl1*, and EpSC+LPC lineage gene *Ptn*. Select clusters are indicated. Cluster numbers are indicated across the bottom. The data underlying this figure can be found in the Gene Expression Omnibus database under accession code GSE218156.
(TIF)

**S5 Fig. ALDH1A1 and FOXA2 expression overlaps in CO mature GE.** Representative ALDH1A1 and FOXA2 IHC in CO adjacent uterine sections ($n$ = 4–6 mice per group).
(TIF)

**S6 Fig. OLFM4 and pAKT staining in DES-exposed uteri.** (A) Representative OLFM4 IHC in CD-1 12-month-old DES uteri ($n$ = 4–6 mice per group). (B) Representative OLFM4 and pAKT IHC in adjacent sections from FVBN/J 9-month-old DES uteri ($n$ = 4–6 mice per group). GE and LE are indicated.
(TIF)

**S7 Fig. Conditional epithelial or stromal deletion of ERα prevents DES-induced uterine cancer.** Representative HE staining of uterine tissue sections from 12-month-old mice exposed neonatally to DES (*n* = 4–6 mice per group). (A) *Esr1*-flox/flox (wild type). (B) *Esr1*-flox/flox; *Amhr2*-cre+ (stromal ERα cKO). (C) *Esr1*-flox/flox; *Wnt7a*-cre+ (epithelial ERα cKO).
(TIF)

**S1 Raw Images. Full unedited immunoblots for Fig 7H.** Left: Immunoblot of OLFM4 in uteri from 12-month-old control or DES-treated mice. Each lane has 5 μg uterine protein extract from 1 mouse (*n* = 3 mice per group). Right: Same blot as in left panel reprobed for actin. Molecular weight markers to left of each blot (kD). WT, wild type; St cKO, stromal ERα conditional knockout; Ep cKO, epithelial ERα conditional knockout.
(TIF)

**S1 Table. Single-cell RNA-seq global data.**
(XLSX)

**S2 Table. Spatial transcriptomics data.**
(XLSX)

**S3 Table. Spatial transcriptomics stromal cell data.**
(XLSX)

**S4 Table. Integrated CO and DES epithelial cell clusters.**
(XLSX)

**S5 Table. CO and DES scRNA-seq non-integrated epithelial cell clusters.**
(XLSX)

**S6 Table. Identification of CO epithelial cell subgroups.**
(XLSX)

**S7 Table. Slingshot analysis of CO glandular epithelium.**
(XLSX)

**S8 Table. Slingshot analysis of CO luminal epithelium.**
(XLSX)

**S9 Table. Characterization of CO glandular progenitor cells, luminal progenitor cells, and epithelial stem cells.**
(XLSX)

**S10 Table. Slingshot analysis of CO glandular progenitor cells, luminal progenitor cells, and epithelial stem cells.**
(XLSX)

**S11 Table. Uterine cancer (DES cluster 5 DEGs) overlapped with Pten conditional knock out prostate tumors versus normal prostate.**
(XLSX)

**S12 Table. Key resources table.**
(DOCX)

**S13 Table. Antibodies used in the current study.**
(XLSX)

## Acknowledgments

We thank Ciro Amato, Humphrey Yao, and Guang Hu (NIEHS) for critical review of the manuscript. We are grateful to Franco DeMayo (NIEHS) for providing the Amhr2-cre mouse line and Ken Korach (NIEHS) for providing the Wnt7a-cre and Esr1-floxed mouse lines.

## Author Contributions

**Conceptualization:** Wendy N. Jefferson, Carmen J. Williams.

**Data curation:** Brian N. Papas.

**Formal analysis:** Brian N. Papas.

**Funding acquisition:** Carmen J. Williams.

**Investigation:** Elizabeth Padilla-Banks, Wendy N. Jefferson, Alisa A. Suen, Xin Xu, Diana V. Carreon, Cynthia J. Willson, Erin M. Quist.

**Methodology:** Elizabeth Padilla-Banks, Wendy N. Jefferson, Brian N. Papas, Alisa A. Suen, Xin Xu, Carmen J. Williams.

**Project administration:** Carmen J. Williams.

**Resources:** Carmen J. Williams.

**Supervision:** Carmen J. Williams.

**Visualization:** Wendy N. Jefferson, Carmen J. Williams.

**Writing – original draft:** Elizabeth Padilla-Banks, Wendy N. Jefferson, Carmen J. Williams.

**Writing – review & editing:** Wendy N. Jefferson, Brian N. Papas, Alisa A. Suen, Xin Xu, Diana V. Carreon, Cynthia J. Willson, Erin M. Quist, Carmen J. Williams.

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
