## [Editor Report · Decision Letter 0]

20 Mar 2023

Dear Carmen, 

Thank you for submitting your manuscript entitled "Developmental endocrine disruption in mice causes uterine adenocarcinoma mediated by Wnt/β-catenin and PI3K/AKT signaling" for consideration as a Research Article by PLOS Biology.

Your manuscript has now been evaluated by the PLOS Biology editorial staff as well as by an academic editor with relevant expertise and I am writing to let you know that we would like to send your submission out for external peer review.

Once your full submission is complete, your paper will undergo a series of checks in preparation for peer review. After your manuscript has passed the checks it will be sent out for review. To provide the metadata for your submission, please Login to Editorial Manager (https://www.editorialmanager.com/pbiology) within two working days, i.e. by Mar 22 2023 11:59PM.

Kind regards,

Luke

Lucas Smith, Ph.D.

Associate Editor

PLOS Biology

lsmith@plos.org

---

## [Decision Letter · Decision Letter 1]

10 May 2023

Dear Carmen,

Thank you for your patience while your manuscript "Developmental endocrine disruption in mice causes adenocarcinoma mediated by Wnt/β-catenin and PI3K/AKT signaling" was peer-reviewed at PLOS Biology. It has now been evaluated by the PLOS Biology editors, an Academic Editor with relevant expertise, and by several independent reviewers. 

In light of the reviews, which you will find at the end of this email, we would like to invite you to revise the work to thoroughly address the reviewers' reports.

As you will see below, the reviewers find the study interesting and a nice contribution to the field, however they have also raised a number of points to further refine and strengthen the study. Given the extent of revision needed, we cannot make a decision about publication until we have seen the revised manuscript and your response to the reviewers' comments. Your revised manuscript is likely to be sent for further evaluation by all or a subset of the reviewers.

**IMPORTANT - SUBMITTING YOUR REVISION**

*Re-submission Checklist*

*Published Peer Review*

*PLOS Data Policy*

*Blot and Gel Data Policy*

Sincerely,

Luke

Lucas Smith, Ph.D.

Associate Editor

PLOS Biology

lsmith@plos.org

REVIEWS:

Reviewer #1: In the current manuscript, the authors applied scRNAseq and spatial transcriptomics to map the differentiation of endometrial epithelia, and reveal the developmental exposure of DES as a big risk of adenocarcinoma progress in adult caused by dysregulation of uterine epithelial differentiation. The findings are really novel and advancing our understanding of stem cell and cancer biology in the uterus. Here are some minor comments for authors to revise the manuscript:

1. A published paper (Reference 17) used lineage tracing to identify the differentiation of epithelial stem cells at a cellular level, in which the stem cell population is located in the intersection area between luminal and glandular epithelia. Here, the authors applied scRNAseq and spatial transcriptomics to further define this differentiation process by mapping the dynamic changes of genes at each stage during the whole epithelial lineage differentiation as well as their location. This map of epithelial differentiation serves as the guidance of cyclical endometrial regeneration, a foundation of uterine health and disease including cancer. This finding is a significant advance to the reproductive biology field. However, the current title of this manuscript does not reflect this key point. A better title is needed to cover both differentiation of epithelial stem cells and cancer formation caused by developmental exposure of DES. 

2. Figure 3G labeling was missing. 

3. Figure 7 legend was mixed in the discussion part. 

4. A specific marker for luminal epithelia in the mouse uterus is lacking currently, it will be a big contribution to the field if authors can further analyze the sequencing data to define a specific marker of LE in the future studies. This comment is just for the future study. 

Reviewer #2: Padilla-Banks et al. use single-cell transcriptomics to characterize an endometrial cancer mouse model based upon diethylstilbesterol (DES) exposure. Overall, the impact of the paper is high given both the sharp increase in endometrial cancer in women and lack of in vivo models that recapitulate endometrial carcinogenesis. Of particular importance is the links identified between DES exposure and known oncogenic and tumor suppressor pathways in endometrial cancer, including PI3K pathway activation with developmental estrogen exposure. Therefore, I support consideration at PLOS Biology and have the following specific comments.

1) The TP53 mutations and/or loss are common in the uterine serous carcinoma, not endometrioid subtypes. In addition, TP53 mutations often don't co-exist PIK3R1 mutations in human endometrial cancer. Most human endometrial cancers without identifiable molecular alterations are TP53 wild-type. Although this does not change the importance of the findings or translational impact, especially in how it pertains to PI3K pathway activation, these details should be clarified in the text and further discussed, since it's likely the DES-model represents a rare or uncommon form of endometrial cancer. Thus, it may limit the use of the DES-model in the exploration of common forms of endometrial cancer. 

2) The mouse sample sizes used for the single cell studies need to be properly justified. Based on the methods, both control and DES-exposed mouse uteri were pooled from individual mice, then processed for library preps and sequencing. It seems odd to pool samples from individual mice, rather than individually examine each mouse in the treatment groups. It's also unclear if the orthogonal assays were performed on the same set of treated mice or if additional treatment groups were analyzed. Overall, the authors should attempt to show that the molecular mechanisms by which DES-exposure promotes endometrial carcinogenesis are recapitulated in the models described. 

Reviewer #3: This study composed an extensive and deep single-cell (sc) transcriptomic atlas of the adult mouse uterus in control and DES-exposed conditions, and extracts interesting findings from it. The dataset will be very useful for the endometrium research community.

The study presents a bioinformatic tour de force with application of multiple cutting-edge computational tools. It touches on the (further) characterization of epithelial stem and progenitor cells of both luminal and glandular epithelium. Interestingly, cell types and clusters are completely different in the DES-exposed uteri and it took original and elaborate approaches to be capable of comparing control with DES conditions. 

Nevertheless, there are still a number remarks and questions, and addressing (some of) them will further strengthen the study.

Introduction

The reason of exposure to DES could be a bit more situated, in view of its past clinical use and associated severe side effects. 

Results

Line 90: " more than 1.2 billion reads per sample": is this very large number correct?

Line 91: The number of CO cells in the scRNA-seq dataset is double the number of DES cells, while uteri from 4 DES mice were pooled compared to only 2 CO mouse uteri. How do the authors explain this? Due to the loss of stromal cells from DES uteri? Could the cell isolation protocol from DES uteri be further optimized not to lose these cells? Then, stromal cells could be analyzed in the scRNA-seq data without the need for spatial transcriptomics.

Line 119-122: DES mesothelial cell gene expression instead of DES epithelial cell gene expression is shown in Fig. S1A; is this a mistake in the text?

Line 125: Basal cells are mentioned, hence this cell type needs much more explanation here. In addition, it is only mentioned in the Discussion that these cells are normally not present in the uterus. Should be described earlier in the Results sections.

Line 146: Spatial transcriptomic analysis was eventually done for only 1 section per condition. How reproducible are the findings? Sections from additional (independent) mouse uteri should be subjected to spatial transcriptomics and analyses included.

Line 186-188: Can the authors provide potential explanations on how DES exposure results in the altered phenotype (including the disrupted expression pattern of Foxl2), regarding e.g. altered signaling pathways by DES, or increased inflammation in stromal cells?

Line 200-201: Fig. 3A seems contradictory to Fig. 1A. In Fig. 1A, DES and CO epithelial cells largely overlap in the UMAP plots. Hence, they would be expected to have similar transcriptomics/gene expression. However, the "overlapping" CO and DES epithelial cells are not present in Fig. 3A? Please clarify.

Line 226: should be Fig. S3A instead of Fig S2A?

Line 227: Foxa2 is not included here. Why?

Line 238: "The lack of overlap in UMAP locations of CO and DES epithelial cells". Their original UMAP (Fig. 1A) clearly shows large overlap of epithelial cells of both groups. Did the authors consider possible 'batch effects' with variations in sample groups due to technical influences rather than biological differences? Please provide more details on the integration methods used in both experiments.

Line 287: Fig. 3F instead of Fig. 3G

Lines 358 and 461: CON should be CO. The latter should be used consistently, also in all figures.

Line 297: For the Slingshot analysis, how was the starting point chosen? Does this occur in an unbiased manner by the program, or does the researcher indicate this starting point? In the latter case, the trajectory may be biased. Please clarify.

Authors elaborate on the GE clusters and their trajectory, but neglect the LE cells in this respect. What are the DEGs/specific markers of LE, and how is the time trajectory within this population? In addition, authors tentatively indicate GE and LE progenitor cells and epithelial stem cells (e.g. 350-353). What is the relationship between these? Pseudotime trajectories should also be determined here. In the end, functional support for stem and progenitor phenotypes is needed by, for instance, lineage tracing using the newly defined markers. Please add to the Discussion.

Lines 397-401 etc: it is strange that the authors aim at testing the identification of the stem and progenitor cells using their identified marker ALDH1A1 and then show its expression in mature GE. This does not fit at all. Should be better and further explained.

Lines 450-451: Six1 was used as basal cell marker (see Fig. 1D), but here it is used as cancer marker in the non-basal cell DES clusters. This is not comprehensible, and should be clarified. Moreover, from the feature plot (Fig. 6A), Six1 is not "most highly expressed in DES cluster 5" as the authors state. High expression is also present in many other DES clusters. Please clarify and adjust if needed.

Libne 489: Wnt signaling may be driving (contributing to) the DES cancer phenotype. Are there mouse models (e.g. beta-cat mutant) that support this?

Line 492: Fig. 5F � There is no panel F in Fig. 5. Should be Fig. 5E?

Line 503: OLFM4 staining is visible in the cavity of the glands (see Fig. 6D and Fig. 6F). Is this background staining or is this secreted protein? Please clarify.

Line 537-539: it is suggested that the stromal inflammation and oxidative stress influences the epithelium. It could also be possible that the abnormal differentiation of epithelial cells induces stromal inflammation and oxidative stress. Please discuss. Moreover, a graphical abstract of the main findings would be clarifying for the reader.

Line 546-547: it is strange that here (Fig. 6H) OLFM4 immunoblotting is used and not staining of sections. It reads as a major gap that no uterine sections of these genetic mouse models are shown. Does histology clearly show the absence of cancer? These analyses should be added.

Line 588-590: If the basal cells do not contribute to the cancer phenotype, what do the authors suggest to be their contribution to the DES phenotype?

Figures:

Line 110: Fig. 1C comment: Top 25 (said in legend) versus Top 50 (shown in figure)

In Fig. 1A, green is used for epithelial cells, in Fig. 1C, green is used for stroma. Could be made more uniform.

Fig. 1C: What does the title "DES" refer to? The UMAP and heatmap plots show both DES and CO.

Fig. 1C left: Epithelial markers are much lower in DES epithelial cluster compared to CO epithelial cluster. However, in Fig. 1D KRT18 expression appears equally high between CO and DES. Please provide a scalebar in Fig. 1D to show the relative expression between the two groups.

Fig. 1A and 2: CON should be replaced by CO (as used in text).

Fig. 2A & C: In our opinion the tissue section used for the CO and DES condition are not of equal quality and are therefore not really comparable. For example, the tissue section used for the CO condition shows a nice, continuous muscle layer whereas the DES tissue section shows a discontinuous muscle layer. Should be explained.

Fig. 3B, F, Fig. 4B: Include scalebars of feature plots to enable reliable comparison of expression levels between CO and DES.

Fig. 3E: FOXA2 signal in not really clear in the LE (is very weak). Authors should provide better stainings/pictures.

---

## [Decision Letter · Decision Letter 2]

20 Jul 2023

Dear Carmen,

Thank you for your patience while we considered your revised manuscript "Developmental estrogen exposure in mice disrupts uterine epithelial cell differentiation and causes adenocarcinoma mediated by Wnt/β-catenin and PI3K/AKT signaling" for consideration as a Research Article at PLOS Biology - and I apologize again for the delay in sending you our decision. Your revised study has now been evaluated by the PLOS Biology editors, the Academic Editor and by two of the original reviewers, whose comments are appended at the end of this email. As you will see, both Reviewers 2 and 3 are fully satisfied by the revision and so we are likely to accept your study. However, while we enthusiastic about the manuscript, overall, the Academic Editor has raised some concerns with the presentation of the piece, that we think should be addressed before publication.

The Academic Editor has commented that while the revision has addressed the reviewer requests, the results still do not point to a single, fully worked out mechanism. Moreover, the Academic Editor has raised the concern that, as written, it is hard to glean clear take-home lessons from the manuscript, noting that the conclusions are still somewhat speculative, and the manuscript is rather complex (for example, the Results and Discussion are quite long, and the graphical abstract is somewhat confusing), so readers may not be able to determine a main finding of the work. 

Before we can accept your study, we think think the manuscript should be carefully edited to address these comments and to make the study more accessible to a general readership. Specifically, the main findings of the work should be further clarified, the results and discussion should be streamlined and shortened and we think that the graphic abstract should either be removed, or edited for clarity. 

Editorially, we think that the study is somewhat in between the scope of our “Research Article” and “Resource” format. However, after some discussion, we end up leaning towards publishing your manuscript as a "Resource", as we think would address some of the Academic Editor’s concerns (and we note the reviewers have commented on the value of the datasets previously). We therefore request that you change the article type to a "Method and Resource" article type. This will likely require reworking the abstract and title a bit, and perhaps reworking some of the sections to highlight the value of the datasets themselves (although resources can also provide interesting new biological insights, as yours does, and so we dont think you need to remove the discussion of your main findings). 

For more information on these article types, see here: https://journals.plos.org/plosbiology/s/what-we-publish

In light of the Academic Editor's comments and the positive reviews, we are pleased to offer you the opportunity to address these comments in a revision that we anticipate should not take you very long. We will then assess your revised manuscript and your response to the reviewers' comments with our Academic Editor aiming to avoid further rounds of peer-review, if possible.

**IMPORTANT: As you make these changes, we also request that you address the following editorial requests: 

1) TITLE: If you agree with pulishing your piece as a Resource, we would suggest the title be changed to reflect that. For example, we suggest it be changed to something like: 

"System-level analysis of uterine epithelial tissue structure and differentiation in normal conditions and after developmental estrogen exposure"

...But Please feel free to refine this further. 

2) ETHICS STATEMENT: Please update the ethics statement, in your methods section, to include the specific national or international regulations/guidelines to which your animal care and use protocol adhered. Please note that institutional or accreditation organization guidelines (such as AAALAC) do not meet this requirement.

3) DATA: Thank you for depositing the sequencing data generated in your study to GEO database. Can you please provide me with a reviewer token so that I can access the data? (sorry if I missed this somewhere). Please also update the figure legends to reference this data. To each relevant figure legend, you can add a note saying "the data underlying this figure can be found in the Gene Expression Omnibus database under accession code GSE218156"

4) WESTERN BLOT: We require the original, uncropped and minimally adjusted images supporting all blot and gel results reported in an article's figures or Supporting Information files. We will require these files before a manuscript can be accepted so please prepare and upload them now. Please carefully read our guidelines for how to prepare and upload this data: https://journals.plos.org/plosbiology/s/figures#loc-blot-and-gel-reporting-requirements

Please provide the uncropped and unadjusted blot images related to Fig 7H. 

**IMPORTANT - SUBMITTING YOUR REVISION**

*Resubmission Checklist*

*Published Peer Review*

Sincerely,

Luke

Lucas Smith, Ph.D.

Senior Editor

PLOS Biology

lsmith@plos.org

REVIEWS:

Reviewer #2: The authors have address my critiques 

Reviewer #3: Authors answered my questions very well and satisfactorily addressed my concerns. The paper became strengthened through this.

---

## [Editor Report · Decision Letter 3]

12 Sep 2023

Dear Carmen,

Thank you for the submission of your revised Research Article "Developmental estrogen exposure in mice disrupts uterine epithelial cell differentiation and causes adenocarcinoma via Wnt/β-catenin and PI3K/AKT signaling". Your revised manuscript has now been assessed by the PLOS Biology editorial team and by a new Academic Editor, Rocio Rivera. I am pleased to say that we are satisfied by the changes made in this revision, in response to our previous editorial requests, and that we can, in principle, accept your manuscript for publication. 

As noted in our previous correspondences over email, since our last decision letter, the original Academic Editor assigned to your manuscript unfortunately needed to step down. In his/her absence, we discussed your manuscript and revision plan with Dr. Rivera, who agreed to step in as Academic Editor - and after that discussion we ultimately agreed with the arguments in your revision plan, including that the manuscript is suitable for publication at PLOS Biology as a research article, and without the major restructuring requested by the original Academic Editor.

Please note that, while we are happy to editorially accept your study, before we can formally accept and schedule your piece for publication, we will need you to address any remaining formatting and reporting issues. These will be detailed in an email you should receive within 2-3 business days from our colleagues in the journal operations team; no action is required from you until then. 

**IMPORTANT: As you address any last formatting and reporting requests to come, we also ask that you address the following editorial request, which I think was missed in the most recent revision: 

>>To each figure legend (including supplemental), please add a brief sentence directing readers to where the underlying data for that figure can be found. For example, to each relevant figure legend, you can add a note saying "the data underlying this figure can be found in the Gene Expression Omnibus database under accession code GSE218156"

PRESS

Sincerely, 

Luke

Lucas Smith, Ph.D.,

Senior Editor

PLOS Biology

lsmith@plos.org